# TAZ inhibits glucocorticoid receptor and coordinates hepatic glucose homeostasis in normal physiological states

Simiao Xu[1,2†], Yangyang Liu[2,3†], Ruixiang Hu[2,4†], Min Wang[2,5,6], Oliver Stöhr[2], Yibo Xiong[2], Liang Chen[2,7], Hong Kang[8], Lingyun Zheng[2], Songjie Cai[2,9], Li He[2], Cunchuan Wang[4], Kyle D Copps[2], Morris F White[2], Ji Miao[2,10]*

[1]Division of Endocrinology, Tongji Hospital, Tongji Medical College, Huazhong University of Science and Technology, Branch of the National Clinical Research Center for Metabolic Disease, Wuhan, China; [2]Division of Endocrinology, Boston Children's Hospital, Harvard Medical School, Boston, United States; [3]Cancer Center, Union Hospital, Tongji Medical College, Huazhong University of Science and Technology, Wuhan, China; [4]Department of Gastrointestinal Surgery, The First Affiliated Hospital of Jinan University, Guangzhou, China; [5]Department of Biliary-Pancreatic Surgery, Tongji Hospital, Tongji Medical College, Huazhong University of Science and Technology, Wuhan, China; [6]Department of Pathology, Beth Israel Deaconess Medical Center, Boston, United States; [7]College of Science, Northeastern University, Boston, United States; [8]Department of Systemic Biology, Harvard Medical School, Boston, United States; [9]Transplantation Research Center, Brigham and Women's Hospital, Harvard Medical School, Boston, United States; [10]Department of Pediatrics, Harvard Medical School, Boston, United States

*For correspondence:
ji.miao@childrens.harvard.edu

†These authors contributed equally to this work

Competing interest: The authors declare that no competing interests exist.

**Abstract** The elucidation of the mechanisms whereby the liver maintains glucose homeostasis is crucial for the understanding of physiological and pathological states. Here, we show a novel role of hepatic transcriptional co-activator with PDZ-binding motif (TAZ) in the inhibition of glucocorticoid receptor (GR). TAZ is abundantly expressed in pericentral hepatocytes and its expression is markedly reduced by fasting. TAZ interacts via its WW domain with the ligand-binding domain of GR to limit the binding of GR to the GR response element in gluconeogenic gene promoters. Therefore, liver-specific TAZ knockout mice show increases in glucose production and blood glucose concentration. Conversely, the overexpression of TAZ in mouse liver reduces the binding of GR to gluconeogenic gene promoters and glucose production. Thus, our findings demonstrate that hepatic TAZ inhibits GR transactivation of gluconeogenic genes and coordinates gluconeogenesis in response to physiological fasting and feeding.

## Introduction

The liver plays a critical role in organismal energy homeostasis by regulating diverse biological processes in response to nutrient availability (*Rui, 2014*). During fasting, the activation of hepatic gluconeogenesis is required for the supply of glucose to tissues with a high glucose demand, such as the brain, and for the maintenance of glucose homeostasis, whereas in the fed state gluconeogenesis is suppressed (*McGarry et al., 1973*; *Landau et al., 1996*). Precise control of hepatic gluconeogenesis is crucial for normal physiology, and a failure to suppress hepatic gluconeogenesis postprandially contributes to hyperglycemia in insulin resistance and diabetes (*Magnusson et al., 1992*).

Glucagon and glucocorticoids (GCs; in humans, cortisol; in mice, corticosterone; synthesized in the adrenal cortex) are hormones that are secreted during fasting and promote hepatic gluconeogenesis at multiple levels, including via gene transcription. Glucocorticoid receptor (GR, encoded by the *NR3C1* gene) is a member of the nuclear receptor super-family (*Hollenberg et al., 1985*) that is a key transcriptional regulator of gluconeogenic gene expression in the fasting state (*Heitzer et al., 2007*). GR not only directly responds to increases in GC concentration by activating gluconeogenic gene transcription, but also plays a permissive role in the glucagon-mediated transcriptional control of these genes. Therefore, the deletion of hepatic GR leads to fasting hypoglycemia (*Opherk et al., 2004*), while adrenalectomy abrogates the induction of gluconeogenesis by fasting, glucagon, cAMP, or epinephrine in rodents (*Exton et al., 1972*; *Winternitz et al., 1957*). Similarly, a single dose of a GR antagonist is sufficient to reduce hepatic glucose output in healthy humans (*Garrel et al., 1995*).

GR transactivation of gluconeogenic genes involves a series of molecular events (*Heitzer et al., 2007*). Upon the binding of a ligand (a synthetic agonist or an endogenous GC) (*Kuo et al., 2015*), GR undergoes conformational changes, translocates to the nucleus, dimerizes, and binds to glucocorticoid response elements (GREs) in the promoters of key gluconeogenic genes, such as phosphoenolpyruvate carboxykinase 1 (*PCK1*) and glucose-6-phosphatase catalytic subunit (*G6PC*) (*Kuo et al., 2015*; *Vander Kooi et al., 2005*), which encode the rate-limiting and final enzymes of gluconeogenesis, respectively (*Mutel et al., 2011*; *She et al., 2000*). This activation mechanism differs from that of the transrepression of inflammatory genes by GR, which involves the tethering of monomeric GR to DNA-bound proinflammatory transcription factors (*Reichardt et al., 2001*).

Yes-associated protein 1 (YAP) and transcriptional co-activator with PDZ-binding motif (TAZ) are downstream effectors of the Hippo pathway (*Patel et al., 2017*). Inhibition of the Hippo pathway activates YAP and TAZ, and they co-activate TEA domain (TEAD) transcription factors within the nucleus, which induce the expression of genes involved in cellular proliferation (*Patel et al., 2017*). However, YAP and TAZ also play other roles, mediated by interactions with diverse transcription factors (*Piccolo et al., 2014*). Although YAP and TAZ share nearly 50% amino acid sequence similarity, the two proteins have distinct functions that are exerted via interactions with different transcription factors (*Piccolo et al., 2014*; *Plouffe et al., 2018*). For example, YAP activates epidermal growth factor receptor 4 (ErbB4) and tumor suppressor p73 (*Komuro et al., 2003*; *Lapi et al., 2008*), whereas TAZ specifically interacts with peroxisome proliferator-activated receptor gamma (PPARγ) and T-box transcription factor (TBX5) (*Hong et al., 2005*; *Murakami et al., 2005*). In addition, although TAZ co-activates TEADs, it also acts as a repressor: the binding of TAZ to PPARγ inhibits the PPARγ-induced differentiation of mesenchymal stem cells into adipocytes (*Hong et al., 2005*). Thus, YAP and TAZ have distinct roles in gene promoter-specific transcriptional regulation.

We previously reported that YAP integrates gluconeogenic gene expression and cell proliferation, which contributes to its tumorigenic effects (*Hu et al., 2017*). However, hepatocyte-specific deletion of YAP in normal mice has little effect on gluconeogenic gene expression (*Hu et al., 2017*). Whereas YAP is primarily expressed in cholangiocytes in normal mouse liver (*Yimlamai et al., 2014*), we show here that TAZ is abundantly expressed in pericentral hepatocytes and that its expression is markedly reduced by fasting. In accordance with these data, we show that TAZ, but not YAP, interacts with GR to inhibit the GR transactivation of gluconeogenic genes, thereby coordinating hepatic glucose production with physiological fasting and feeding in normal mouse liver.

## Results

### Fasting and feeding alter hepatic TAZ protein level

To determine whether TAZ plays a role in hepatic metabolic regulation under normal physiological conditions, we measured hepatic TAZ expression in mice that had been fed ad libitum or fasted for 24 hr. As expected, fasting increased the hepatic mRNA expression of genes encoding the key gluconeogenic enzymes *Pck1* and *G6pc,* but it inconsistently affected the expression of canonical TAZ and YAP target genes involved in cell proliferation (*Patel et al., 2017*), including connective tissue growth factor (*Ctgf*) and cysteine-rich angiogenic inducer 61 (*Cyr61*) (*Figure 1—figure supplement 1A and B*). Hepatic TAZ protein level was reduced by >50% after 24 hr of fasting, relative to ad libitum feeding (*Figure 1A and B*). The phosphorylation of TAZ at serine 89, which is mediated by the Hippo pathway core kinase large tumor suppressor kinase (LATS)1/2 (*Lei et al., 2008*), was commensurate with the

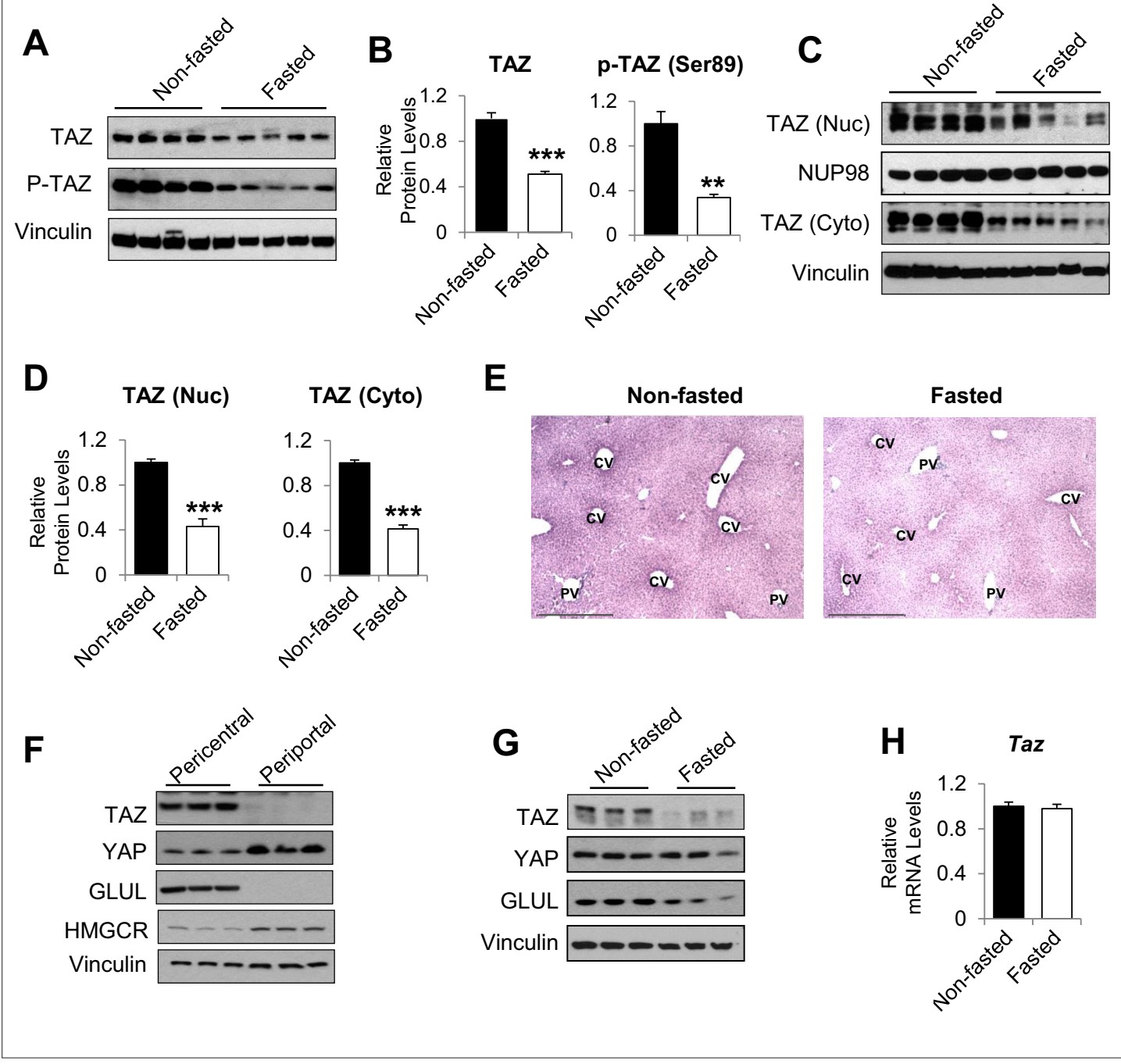

**Figure 1.** humanTAZ protein is regulated by fasting and feeding in hepatocytes. 8- to 12-week-old C57BL/6J mice were ad libitum-fed (non-fasted) or fasted for 24 hr (fasted). (**A–D**) Hepatic proteins were measured by immunoblotting whole-cell lysates (**A**; quantified results are shown in **B**) or nuclear (Nuc) or cytoplasmic (Cyto) extracts (**C**; quantified results are shown in **D**). (**E**) Immunohistochemical staining for TAZ in the livers of mice fed ad libitum or fasted for 24 hr (CV, central vein; PV, portal vein; scale bar, 500 μM). (**F, G**) Protein levels were measured by immunoblotting lysates of pericentral and periportal mouse hepatocytes (**F**) or hepatocytes isolated from mice that were ad libitum-fed or fasted for 24 hr (**G**). (**H**) mRNA expression of hepatic mRNA expression of hepatic *Taz* was measured using real-time qRT-PCR. Data are means and SEMs; control values were set to 1; n = 6. Representative results of 2–3 independent experiments are shown. Data were analyzed by unpaired Student's *t*-test; **p<0.01, ***p<0.001.

The online version of this article includes the following figure supplement(s) for figure 1:

**Figure supplement 1.** Hepatic gene expression in ad libitum-fed and fasted (24 hr) mouse livers.

**Figure supplement 2.** Validation of immunohistochemical staining for TAZ in control and L-TAZ KO liver sections.

**Figure supplement 3.** mRNA and protein expression of TAZ in primary mouse hepatocytes.

reduction in total TAZ protein (*Figure 1A and B*). Both nuclear and cytoplasmic TAZ proteins shared this regulation (*Figure 1C and D*).

Immunohistochemistry confirmed that TAZ was abundantly expressed in both the nuclear and cytoplasmic compartments of mature hepatocytes and that its expression was reduced by fasting (*Figure 1E*). The antibody used was validated using liver sections from liver-specific TAZ knockout (L-TAZ KO) mice (which were generated by crossing TAZ floxed mice with *Albumin*-Cre mice) (*Figure 1—figure supplement 2*). Interestingly, TAZ showed zonal expression, with the highest protein levels being found in the pericentral (or perivenous) hepatocytes (*Figure 1E*). This zonal expression negatively correlated with the expression and activity of gluconeogenic genes in the liver lobule (*Jungermann and Thurman, 1992*). Immunoblotting of isolated pericentral and periportal hepatocytes confirmed that TAZ was primarily expressed in glutamine synthetase (GLUL)-expressing pericentral hepatocytes, whereas YAP is primarily expressed in periportal hepatocytes (*Figure 1F*). These results are consistent with the higher *Taz* mRNA expression in pericentral than periportal mouse hepatocytes, revealed by single-cell RNA analysis (*Halpern et al., 2017*). In addition, TAZ protein was less abundant in hepatocytes isolated from fasting than ad libitum-fed mice (*Figure 1G*), suggesting that fasting reduces TAZ protein in hepatocytes.

In contrast to TAZ protein levels, *Taz* mRNA was not affected by fasting or feeding (*Figure 1H*), indicating that TAZ is post-transcriptionally regulated by physiological fasting and feeding. These data are consistent with previous findings that TAZ is subject to ubiquitin-mediated degradation (*Huang et al., 2019*). Consistent with this, TAZ protein, but not its mRNA, was induced in mouse primary hepatocyte cultures by supplementation of the medium with a high glucose concentration (25 mM) and 10% fetal bovine serum (FBS), which mimics the fed condition, in a time-dependent manner (*Figure 1—figure supplement 3A and B*).

## Knockdown or knockout of hepatic TAZ induces gluconeogenic gene expression and hyperglycemia in mice

To define the role of TAZ in glucose homeostasis in mouse liver, we acutely knocked down hepatic TAZ using adenoviruses expressing shRNA targeting *Taz* (AdshTAZ) or control shRNA targeting human lamin (AdshCon). Compared with the control virus, administration of AdshTAZ to C57BL/6J mice reduced hepatic TAZ mRNA and protein levels (*Figure 2A and B*). TAZ knockdown did not alter mouse body or epididymal white adipose tissue mass, but it slightly reduced liver mass, without affecting liver histology (*Figure 2—figure supplement 1A–D*). However, TAZ knockdown significantly increased the mRNA expression of hepatic *Pck1* and *G6pc* by six- and twofold (*Figure 2C*), respectively, and PCK1 and G6PC protein levels twofold (*Figure 2D and E*). Consistent with this, knockdown of hepatic TAZ significantly increased the ad libitum-fed and fasting blood glucose concentrations (*Figure 2F*). Mice with hepatic TAZ knockdown also showed larger blood glucose excursions than control mice when challenged with pyruvate, a gluconeogenic substrate (*Figure 2G*). By contrast, the knockdown of TAZ did not alter the levels of YAP or key gluconeogenic factors (e.g., hepatic nuclear receptor alpha [HNF4α], forkhead box O1 [FoxO1], and GR; *Goldstein and Hager, 2015*), suggesting that the inhibition of gluconeogenic gene by TAZ is unlikely to occur via regulation of the protein levels of these key transcription factors (*Figure 2—figure supplement 2*). It also had little effect on the phosphorylation of AKT, FoxO1, and CREB (*Figure 2—figure supplement 2*), or on the plasma concentrations of insulin or glucagon (*Figure 2—figure supplement 3A and B*), hormones that regulate gluconeogenic gene expression.

To validate these results, we used a genetic TAZ deletion model, L-TAZ KO mice (*Taz^{flox/flox}:Albumin*-Cre), in which TAZ is deleted in fetal liver. L-TAZ KO mice were born at Mendelian ratios and showed little difference in body mass from sex and age-matched littermate floxed controls (Flox) (*Figure 2—figure supplement 4A*). The knockout of TAZ was confirmed by immunoblotting lysates prepared from liver and isolated hepatocytes (*Figure 2H and I*). The knockout of TAZ also had no overt effects on liver mass or histology (*Figure 2—figure supplement 4B and C*). Previous studies showed that TAZ deletion in phosphatase tension and homolog (PTEN) knockout mice reduces insulin receptor substrate (IRS) 2 expression in mouse liver (*Jeong et al., 2018*) and that TAZ deletion in muscle reduces IRS1 expression (*Hwang et al., 2019*). In addition, it has been shown that TAZ deletion in white adipose tissue improves insulin sensitivity (*El Ouarrat et al., 2020*). However, L-TAZ KO mice did not have differing levels of IRS1 and 2 expression or insulin sensitivity relative to floxed

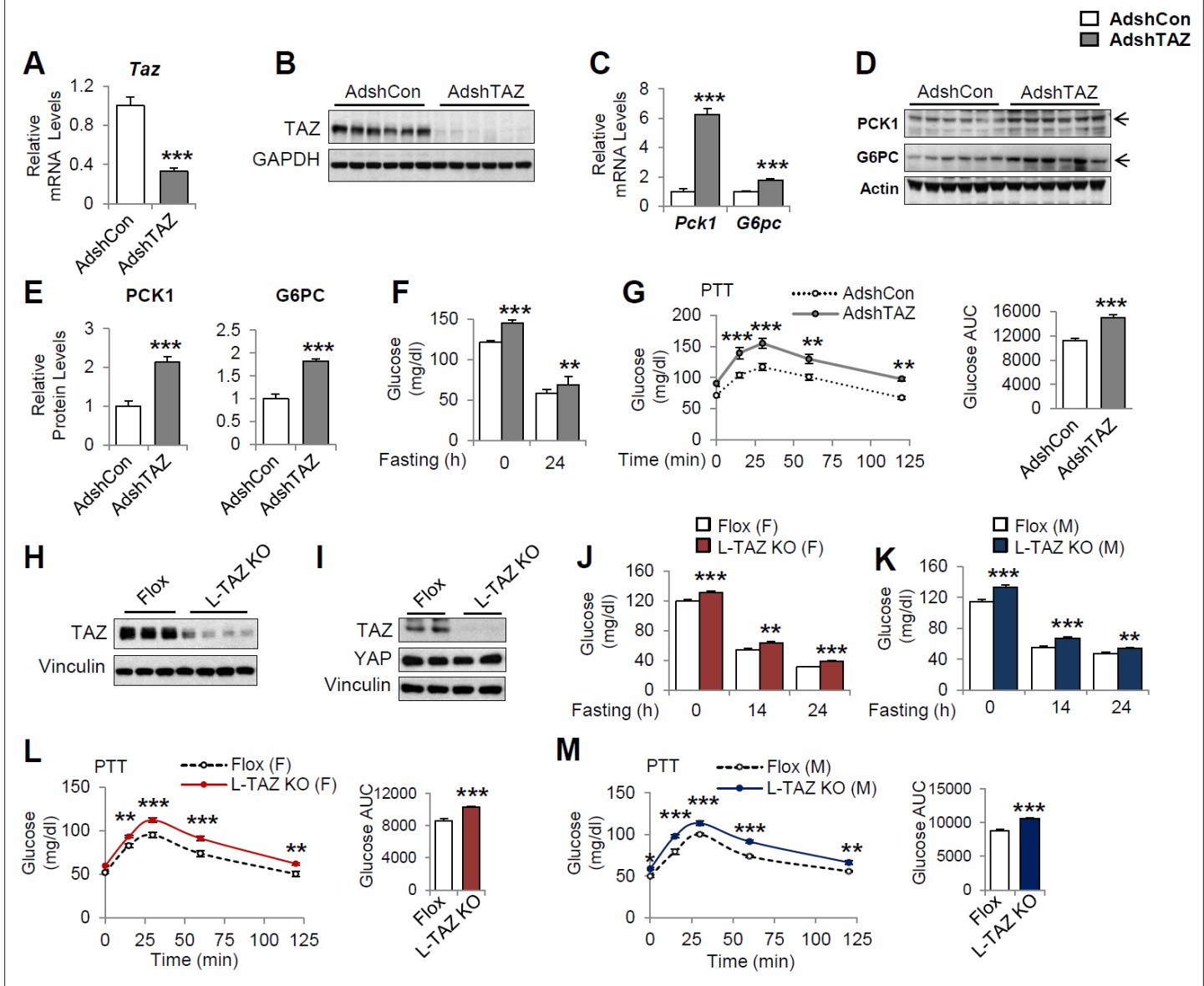

**Figure 2.** Knockdown or knockout of hepatic TAZ increases gluconeogenic gene expression and blood glucose concentrations in mice. (**A–G**) 8- to 12-week-old C57BL/6J mice were administered AdshTAZ or AdshControl (AdshCon), then sacrificed 8 days later in the ad libitum-fed state, when the mRNA expression of hepatic genes (**A, C**) and the corresponding protein levels (**B, D,** and **E** [quantified results of **D**]) were measured. (**F**) Ad libitum-fed and fasting blood glucose concentrations were measured. (**G**) Six days after adenoviral injection, the mice underwent pyruvate tolerance testing (PTT) (left) and the areas under the curves (AUCs) were calculated (right); p<0.001 AdshTAZ versus AdshCon. (**H–M**) 10- to 12-week-old, age- and sex-matched L-TAZ KO and floxed (Flox) control littermates were studied. (**H, I**) TAZ protein was measured in whole liver cell lysates (**H**) or isolated hepatocytes (**I**) by immunoblotting. (**J, K**) Ad libitum-fed and fasting blood glucose concentrations were measured. (**L, M**) Mice underwent PTT; p<0.001 L-TAZ KO versus Flox. Data are means and SEMs; control values were set to 1 (**A, C, E**); n = 7–10. Representative results of 2–3 independent experiments are shown. Data were analyzed by unpaired Student's *t*-test (**A, C, E–F, G** [right], **J–K, L–M** [right]) and two-way ANOVA (**G** [left] and **L–M** [left]); *p<0.05, **p<0.01, ***p<0.001.

The online version of this article includes the following figure supplement(s) for figure 2:

**Figure supplement 1.** Information on mice administered AdshTAZ or AdshCon.

**Figure supplement 2.** TAZ knockdown has no effects on protein expression of key gluconeogenic factors and insulin and glucagon signaling in mice.

**Figure supplement 3.** TAZ knockdown has no effects on plasma insulin and glucagon concentrations in the ad libitum-fed state.

**Figure supplement 4.** Information on L-TAZ KO and flox mice.

**Figure supplement 5.** L-TAZ KO has no effects on insulin sensitivity.

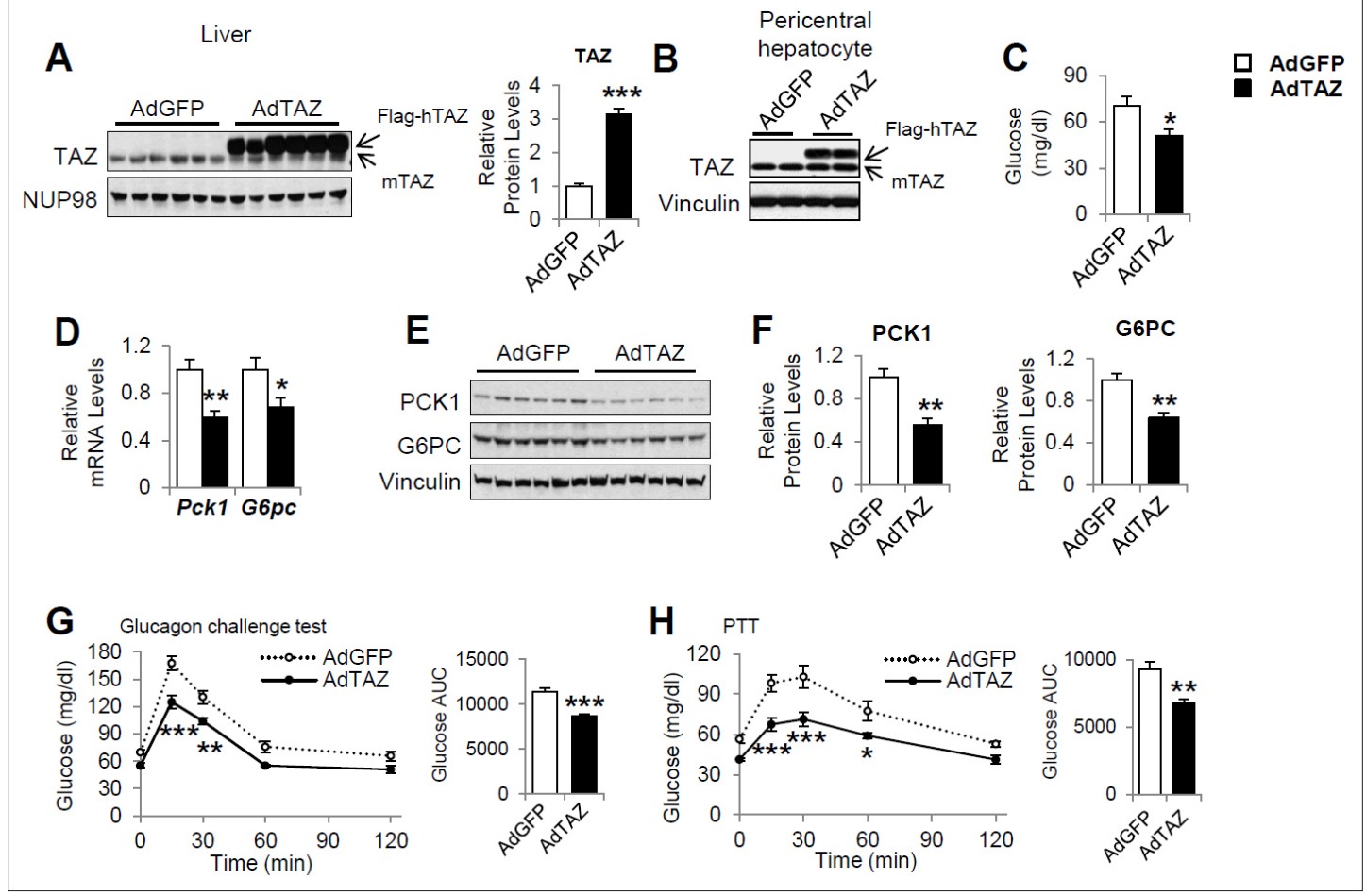

**Figure 3.** Overexpression of hepatic TAZ inhibits gluconeogenic gene expression and reduces blood glucose concentration in mice. 8- to 12-week-old C57BL/6J mice were administered AdGFP or AdTAZ, then sacrificed 5 days later, after a 24 hr fast, when the hepatic protein levels in nuclear extracts (**A** [representative image, left; quantified results, right]), isolated pericentral hepatocytes (**B**), or whole-cell lysates (**E, F** [quantified results of **E**]) were measured. Blood glucose (**C**) and mRNA expression of hepatic genes (**D**) were measured. (**G, H**) Five days after adenoviral injection, the mice underwent glucagon challenge (**G**) or pyruvate tolerance testing (PTT) (**H**); p<0.001 L-TAZ KO versus Flox. Data are means and SEMs; control values were set to 1 (**A, D, F**); n = 5–10. Representative results of 2–3 independent experiments are shown. Data were analyzed by unpaired Student's *t*-test (**A, C, D, F, G** [right], **H** [right]) and two-way ANOVA (**G** [left], **H** [left]); *p<0.05, **p<0.01, ***p<0.001.

The online version of this article includes the following figure supplement(s) for figure 3:

**Figure supplement 1.** Immunohistochemical staining for TAZ in livers of mice administered AdTAZ or control AdGFP and fasted for 24 hr.

**Figure supplement 2.** Information on mice administered AdTAZ or control AdGFP.

controls (*Figure 2—figure supplement 5A-C*). Nonetheless, similar to the aforementioned AdshTAZ-treated mice, both male and female L-TAZ KO mice had higher ad libitum-fed and fasting blood glucose concentrations (*Figure 2J and K*). These knockout mice also showed a larger increase in blood glucose during pyruvate tolerance testing (PTT) (*Figure 2L and M*). Taken together, these data show that deletion of hepatic TAZ produces a fasting-like state in the liver, in which gluconeogenic gene expression and glucose production are induced.

## Hepatic overexpression of TAZ reduces gluconeogenic gene expression and blood glucose in mice

To determine whether overexpression of TAZ is sufficient to inhibit the expression of gluconeogenic genes in the fasting state, we constructed an adenovirus expressing flag epitope-tagged human TAZ (AdTAZ), which expresses a slightly larger protein than endogenous mouse TAZ (*Figure 3A and B*). Although endogenous TAZ protein primarily expresses in pericentral hepatocytes,

immunohistochemistry confirmed the expression of AdTAZ in both pericentral and periportal hepatocytes (*Figure 3—figure supplement 1*). Immunoblotting revealed that in pericentral hepatocytes the level of exogenously expressed TAZ was comparable to endogenously expressed TAZ (*Figure 3B*). Compared with the administration of control virus expressing green fluorescent protein (AdGFP), AdTAZ administration caused relative hypoglycemia during fasting (*Figure 3C*), which was accompanied by lower mRNA expression and protein levels of hepatic PCK1 and G6PC (*Figure 3D and E*). TAZ overexpression did not affect mouse body or white adipose mass, but modestly increased liver mass, without affecting liver histology (*Figure 3—figure supplement 2A–D*). TAZ overexpression also blunted the rise in blood glucose caused by the injection of glucagon or pyruvate (*Figure 3G and H*), without affecting the protein levels of key gluconeogenic factors, the phosphorylation of AKT or CREB, or plasma insulin or glucagon concentrations (*Figure 3—figure supplement 2E and F*). Taken together, these data indicate that the overexpression of TAZ in the fasting state mimics the fed state and is sufficient to inhibit hepatic gluconeogenic gene expression and glucose production.

## The regulation of gluconeogenic gene expression by TAZ is hepatocyte-autonomous

Infection of mouse primary hepatocytes with AdshTAZ reduced both TAZ mRNA expression and protein level (*Figure 4A*) and markedly increased the glucagon and dexamethasone (Dex)-induced mRNA expression of *Pck1* and *G6pc* (*Figure 4B and C*). TAZ knockdown also increased glucose secretion into the medium by glucagon-stimulated hepatocytes (*Figure 4D*), which demonstrates the functional importance of these changes in gene expression. Conversely, the infection of mouse primary hepatocytes with AdTAZ increased TAZ protein levels threefold compared with infection with control AdGFP (*Figure 4E*) and substantially inhibited the glucagon and Dex-induced mRNA expression of *Pck1* and *G6pc* (*Figure 4F and G*). Consistent with these data, AdTAZ infection also reduced glucose production after glucagon stimulation (*Figure 4H*). In addition to gluconeogenic genes, AdTAZ blunted the effect of Dex on the induction of the mRNA expression of glycogen synthase 2 (*Gys2*), a GR target gene involved in glycogen metabolism (*Bose et al., 2016*; *Figure 4—figure supplement 1*).

In addition, knockdown or overexpression of TAZ did not alter the protein levels of YAP or gluconeogenic transcription factors (*Figure 4—figure supplement 2A and B*). Given that insulin and glucagon are major regulators of gluconeogenic genes, we determined whether TAZ regulates their action in hepatocytes and found that it had no effects on either the basal or stimulated phosphorylation of CREB or AKT in hepatocytes treated with glucagon or insulin, respectively (*Figure 4—figure supplement 3A-D*), suggesting that TAZ is unlikely to directly affect gluconeogenic gene expression by altering the cellular sensitivity to these hormones in hepatocytes.

## TAZ inhibits the GR transactivation of gluconeogenic genes

We next determined the molecular mechanisms by which TAZ inhibits gluconeogenic gene expression. Because TAZ is not known to bind directly to gluconeogenic gene promoters, we determined whether the transcriptional effects of TAZ are dependent on other factors in cell culture. Consistent with the results of previous studies, co-transfection of HepG2 cells with vectors expressing GR, HNF4α, or PGC1α, followed by Dex treatment, induced human *G6PC* and *PCK1* luciferase reporter activity (*G6PC*-Luc and *PCK1*-Luc, respectively) (*Figure 5A*, *Figure 5—figure supplement 1A*). Overexpression of TAZ in these cells inhibited the activity of the *G6PC* and *PCK1* promoters by 50% (*Figure 5A*, *Figure 5—figure supplement 1A*), indicating that TAZ suppresses transcription of these genes. To determine whether the Hippo pathway and TEADs are required for these effects of TAZ, we utilized TAZS89A (*Lei et al., 2008*) and TAZS51A (*Zhang et al., 2009*) mutants, which abolish TAZ phosphorylation by LATS1/2 and cannot interact with TEADs, respectively. Both mutants inhibited *G6PC*-Luc to a similar extent, compared to wild-type TAZ (*Figure 5A*), suggesting that the Hippo pathway and TEADs are not required. Conversely, the transfection of cells with a vector expressing an shRNA targeting human TAZ reduced TAZ expression (*Figure 5—figure supplement 1B*) and increased the activity of *G6PC*-Luc and *PCK1*-Luc by >40% (*Figure 5B*, *Figure 5—figure supplement 1C*).

To identify the molecular target of TAZ, we constructed a luciferase reporter containing three repeats of a canonical GR response element (inverted hexameric half-site motifs, separated by a three-base-pair spacer) (*GRE*-Luc) (*Meijsing et al., 2009*). Similar to our findings using *G6PC*-Luc and

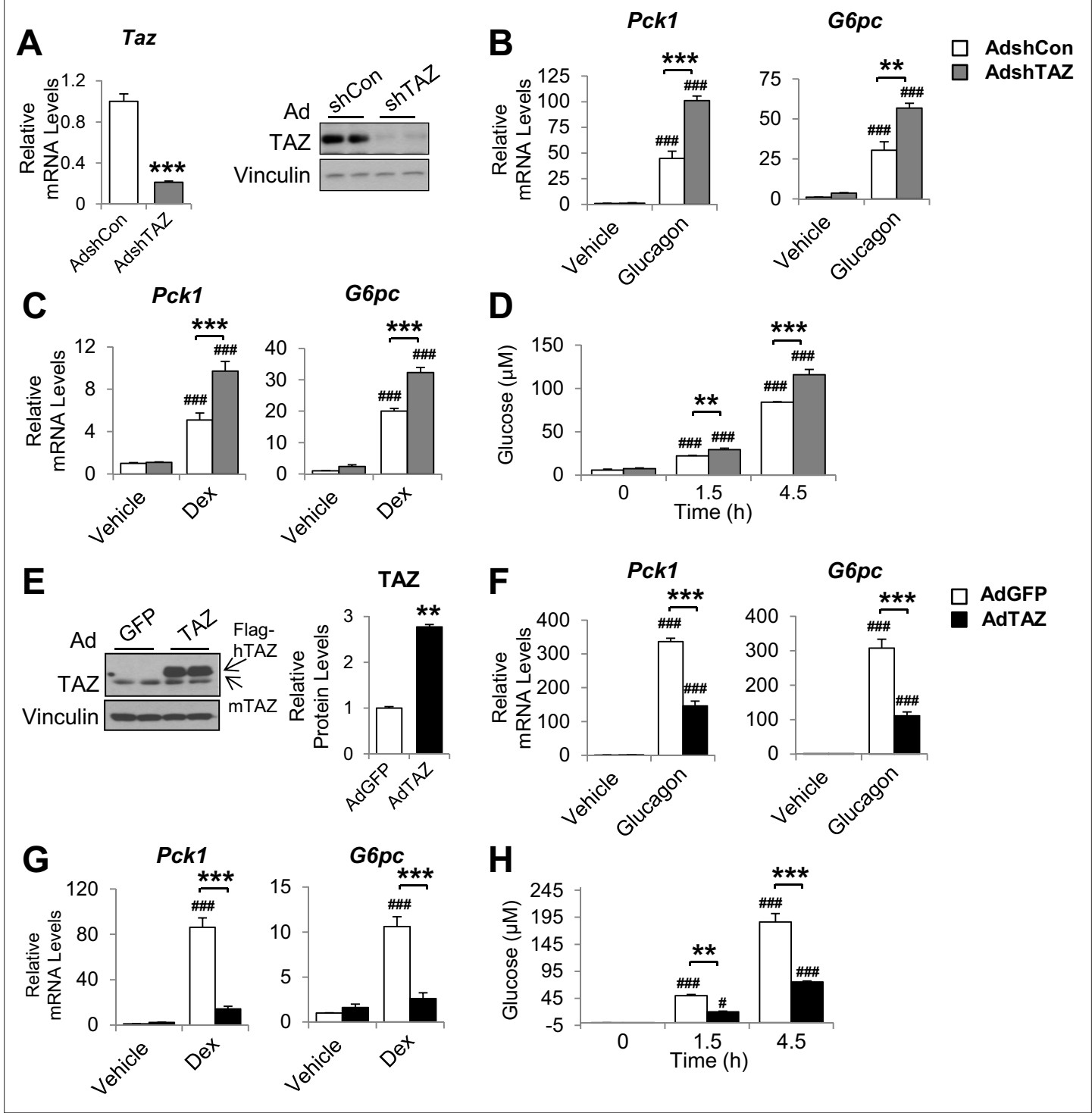

**Figure 4.** The inhibition of gluconeogenic gene expression by TAZ is hepatocyte-autonomous. Primary mouse hepatocytes were isolated from 8- to 12-week-old C57BL/6J mice. (**A–D**) To knock down TAZ, cells were infected with AdshTAZ or AdshControl (AdshCon). (**E–H**) To overexpress TAZ, cells were infected with AdTAZ or AdGFP. Cells were treated with glucagon (20 nM) for 3 hr (**B, F**) or dexamethasone (Dex, 100 nM) for 6 hr (**C, G**). (**A** [left]–**C, F, G**) Gene expression was measured using real-time RT-PCR. (**A** [right], **E**) TAZ protein levels were measured by immunoblotting whole-cell lysates and the quantified results of (**E**) are shown on the right. (**D, H**) Glucose production, in the presence of glucagon (20 nM), was assessed by measuring the glucose concentration in the media at the indicated times. Data are means and SEMs of three wells; control values were set to 1. Data were analyzed by unpaired Student's $t$-test (**A, E**) and two-way ANOVA (**B–D** and **F–H**). *$p<0.05$, **$p<0.01$, ***$p<0.001$ versus control adenovirus-treated cells; in (**B–D**) and (**F–H**), #$p<0.05$, ###$p<0.001$ versus similarly treated controls. Representative results of 2–5 independent experiments are shown.

*Figure 4 continued on next page*

**Figure supplement 1.** The effect of TAZ on the induction of glycogen synthase by dexamethasone (Dex).

**Figure supplement 2.** Protein expression in primary mouse hepatocytes in the presence of TAZ knockdown or overexpression.

**Figure supplement 3.** TAZ knockdown or overexpression in primary hepatocytes has no effects on insulin or glucagon signaling.

*PCK1*-Luc, TAZ suppressed Dex-stimulated *GRE*-Luc activity by >50%, whereas knockdown of TAZ increased promoter activity by 40% (*Figure 5C and D*), suggesting that TAZ inhibits GR transactivation.

## The interaction of TAZ with GR is dependent on its WW domain

The WW domain of TAZ has been reported to interact with proteins containing an I/L/PPxY (I, isoleucine; L, leucine; P, proline; x, any amino acid; and Y, tyrosine) motif (*Kanai et al., 2000*; *Hong et al., 2005*). Moreover, an I/LPxY motif found in the ligand-binding domain of GR (*Giguère et al., 1986*) is highly conversed across species, including in humans, mice, and rats (*Figure 5—source data 1*). Co-immunoprecipitation (co-IP) assay revealed that TAZ, but not YAP, was able to interact with GR (*Figure 5E*). These data are consistent with our previous findings that YAP does not interact with PGC1α (*Hu et al., 2017*), a co-activator of GR. A TAZ mutant protein lacking the WW domain (TAZΔWW), but not one lacking the coiled-coil (CC) domain (TAZΔCC), showed much weaker interaction with GR (*Figure 5F*) and was unable to inhibit *GRE*-Luc (*Figure 5G*). Consistent with this, compared with the TAZS89A mutant that suppressed glucagon or Dex-induced activation of *Pck1* and *G6pc* gene expression by >80% in primary mouse hepatocytes, TAZΔWW failed to suppress glucagon or Dex-induced *Pck1* and *G6pc* expression (*Figure 5H–J*). Furthermore, a TAZ mutant consisting of only the WW domain (TAZWW) was sufficient to inhibit *G6PC*-Luc and *GRE*-Luc activity (*Figure 5K and L*), but was unable to activate TEAD, due to the lack of a TEAD-binding domain (*Figure 5—figure supplement 2*), suggesting that the effects of TAZ on gluconeogenic genes can be separated from its effects on proliferative genes.

Conversely, a GR mutant in which the conserved IPKY motif was mutated to alanines (A) (GR4A mutant) interacted with TAZ much more weakly, despite being expressed at a level similar to that of the wild-type GR (*Figure 5M*). In addition, compared with wild-type GR, the GR4A mutant was not subject to TAZ-mediated inhibition of *GRE*-Luc and displayed a greater ability to induce the activities of the *GRE*-Luc and *G6PC*-Luc reporters (*Figure 5N and O*).

## TAZ inhibits the binding of GR to GREs

To understand how the interaction between TAZ and GR inhibits GR transactivation, we determined how the binding of TAZ to GR inhibits GR nuclear localization, dimerization, and binding to promoter GREs. Whereas Dex treatment induced the nuclear accumulation of GR, overexpression of TAZ had little effect on its subcellular distribution in either the absence or presence of Dex (*Figure 5—figure supplement 3A*). Similarly, TAZ did not reduce the quantity of dimeric GR in cells, as shown by immunoblotting after the treatment of cells with a cross-linker (dithiobis [succinimidyl propionate] [DSP]) (*Figure 5—figure supplement 3B*). In addition, the fact that the GR4A mutant can be activated by Dex (*Figure 5N and O*) also suggests that the TAZ-GR interaction does not impair GR nuclear localization, ligand activation, or dimerization.

To evaluate GR binding to the *G6pc* and *Pck1* promoters, we performed chromatin immunoprecipitation (ChIP) assays in primary mouse hepatocytes using control IgG and anti-GR antibodies. Glucagon treatment strongly promoted the binding of GR to the *Pck1* and *G6pc* promoter regions containing GREs ('*Pck1* (GRE)', −376 to −280 and '*G6pc* (GRE)', −215 to −111), but not to a distal region of the *Pck1* promoter lacking GREs ('*Pck1* (Con)', −3678 to −3564). However, the glucagon-induced binding of GR to the *Pck1* or *G6pc* promoters was significantly lower in cells overexpressing TAZ (*Figure 5P*). Taken together, these data suggest that the TAZ-GR interaction limits gluconeogenic gene expression by reducing GR binding to GREs.

To determine whether TAZ might have significant GR-independent effects to reduce hepatic gluconeogenic gene expression, we treated primary mouse hepatocytes with RU486, a well-characterized and potent GR antagonist (*Schulz et al., 2002*). Upon binding to GR, RU486 induces a conformational change that favors the interaction of GR with co-repressors, but stabilizes the binding of dimeric

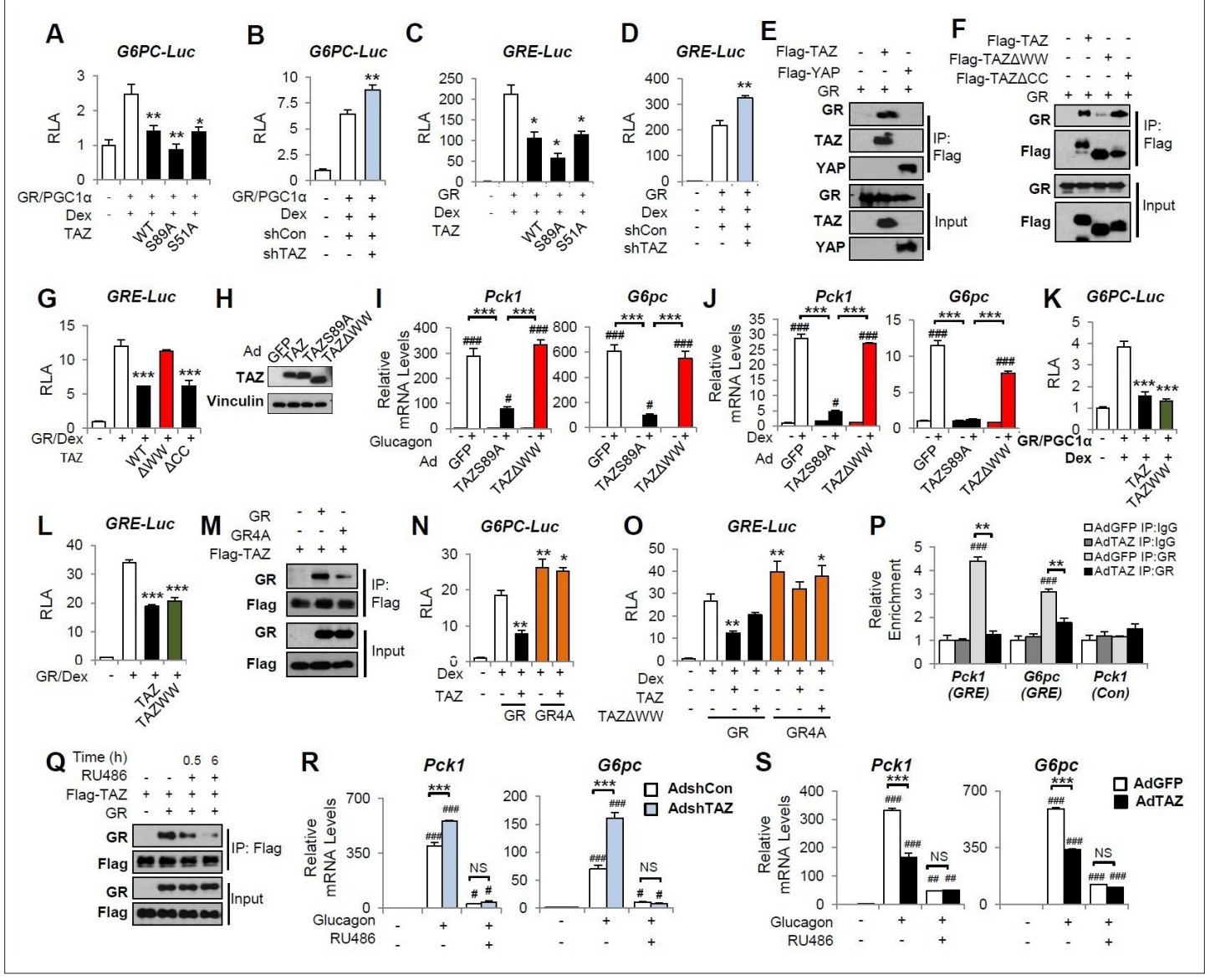

**Figure 5.** TAZ interacts with glucocorticoid receptor (GR) and inhibits the transactivation of gluconeogenic genes by GR. (**A–D, G, K, L, N, O**) HepG2 cells were co-transfected with expression vectors, luciferase reporters, and an internal control (*Renilla*), and treated with dexamethasone (Dex; 100 nM), as indicated. Relative luciferase activity (RLA) is presented after normalization to the *Renilla* activity. (**E, F, M, Q**) 293A cells were transfected with expression vectors, which was followed by immunoprecipitation and immunoblotting, as indicated. (**H–J, R, S**) Primary mouse hepatocytes were infected with adenoviruses, with or without glucagon or Dex treatment. (**R, S**) Cells were treated with RU486 30 min prior to glucagon treatment. Protein levels (**H**) and gene expression (**I, J, R, S**) were measured by immunoblotting and real-time RT-PCR, respectively. (**P**) Chromatin immunoprecipitation (ChIP) assays were performed using an anti-GR antibody or a control IgG. The relative enrichment of GR was assessed using real-time RT-PCR, and primers for the indicated regions of the *Pck1* and *G6pc* genes were performed using an anti-GR antibody or a control IgG. Data are means and SEMs of three or four wells or immunoprecipitation reactions. Representative results of 2–5 independent experiments are shown. Data were analyzed by one-way ANOVA (**A–D, G, K, L, N, O**) and two-way ANOVA (**I, J, P, R, S**). *p<0.05, **p<0.01, ***p<0.001, #p<0.05, ##p<0.01, ###p<0.001. In (**A–D, G, K, L, N, O**), * denotes comparisons with wells treated with GR/Dex or GR/PGC1α/Dex alone; in (**I, J R, S**), # denotes comparisons with controls administered the same viruses; in (**P**), # denotes a comparison with IgG.

The online version of this article includes the following source data and figure supplement(s) for figure 5:

**Source data 1.** A conserved I/LPKY motif is identified in glucocorticoid receptor (GR) from various species.

**Figure supplement 1.** TAZ inhibits *PCK-Luc* activity.

**Figure supplement 2.** Effects of TAZ mutants on TEAD transactivation.

**Figure supplement 3.** TAZ has no effects on glucocorticoid receptor (GR) nuclear localization or the amounts of GR dimer.

GR to GREs (*Schulz et al., 2002*). Interestingly, we found that RU486 blocked the interaction of GR with TAZ (*Figure 5Q*), probably due to the change in GR conformation. As expected, RU486 treatment of hepatocytes reduced glucagon-stimulated *Pck1* and *G6pc* mRNA expression (*Figure 5R and S*), whereas the knockdown of TAZ increased glucagon-stimulated *Pck1* and *G6pc* expression in the absence of RU486. However, this increase was entirely abolished in the presence of RU486 (*Figure 5R*). Similarly, RU486 prevented the inhibitory effects of TAZ overexpression on *Pck1* and *G6pc* expression in hepatocytes (*Figure 5S*). Thus, not only is the TAZ-GR interaction necessary for the reduction in gluconeogenic gene expression, but it is likely the sole mechanism for such a reduction.

## TAZ inhibits GR transactivation in mice

Consistent with the interaction between GR and TAZ identified in cultured cells, GR in the nuclear extracts of wild-type mouse liver could be immunoprecipitated using an antibody targeting endogenous TAZ (*Figure 6A*). To verify that TAZ regulates the association of GR with promoter GREs in mouse liver, we conducted ChIP assays of endogenous gluconeogenic gene promoters. GR bound to regions of the *G6pc* and *Pck1* promoters containing functional GREs (*Figure 6B*). The binding of GR to GRE-containing regions of the *G6pc* and *Pck1* promoters was significantly increased by TAZ knockdown (*Figure 6B*). Moreover, the histone acetylation of *G6pc* and *Pck1* promoter regions containing or near to GREs was increased 50–80% by TAZ knockdown (*Figure 6C*), implying active transcription from the *G6pc* and *Pck1* genes. Conversely, ChIP assays revealed that TAZ overexpression caused a 25–45% reduction in the binding of GR to GREs, histone acetylation, and the binding of the polymerase II subunit (Pol II) to the promoter regions of *G6pc* and *Pck1* (*Figure 6D–F*). However, TAZ bound normally to the promoter of *Ctgf* (a well-described TAZ target gene that is not known to be regulated by GR) in AdTAZ-infected mice, but not to the promoters of *G6pc* or *Pck1* (*Figure 6—figure supplement 1*). These data support a model in which TAZ inhibits GR transactivation by interacting with GR and suppressing GR binding to GREs, rather than a model in which TAZ binding near or within GREs inhibits the binding of GR to gluconeogenic gene promoters.

We next determined whether the inhibition of gluconeogenic gene expression by TAZ requires GR. Because RU486 reduced the TAZ-GR interaction in cultured cells (*Figure 5Q*) and mouse liver nuclear extracts (*Figure 6G*), we determined whether RU486 could abrogate the effects of TAZ on glucose homeostasis in mice. As expected, TAZ knockdown increased mouse blood glucose concentration prior to RU486 injection. However, this increase was completely abolished by RU486 treatment (*Figure 6H*). Similarly, hepatic TAZ knockout increased fasting blood glucose and glucose production after pyruvate administration to mice; and RU486 treatment abolished the effects of TAZ on blood glucose (*Figure 6I*). RU486 is a GR antagonist, and thus it lowered glucose production in floxed mice during a PTT. Similar to its effect on blood glucose concentration, RU486 entirely abolished the effects of hepatic TAZ knockout on glucose production (*Figure 6J*). Consistent with these loss-of-function studies, RU486 also entirely abolished the ability of overexpressed hepatic TAZ to reduce gluconeogenic gene expression, improve pyruvate tolerance, and inhibit gluconeogenic gene expression (*Figure 6K–M*). Moreover, RU486 abolished the ability of TAZ to reduce GR binding to GREs in the *Pck1* or *G6pc* promoters and the amounts of acetylated histones on these promoters (*Figure 6N*). These data confirm that the interaction between GR and TAZ is required for the inhibitory effects of TAZ on hepatic gluconeogenic gene expression in mice.

The binding of dimeric GR to GREs is required for the GR transactivation of gluconeogenic genes, but not GR transrepression of anti-inflammatory genes (*Reichardt et al., 2001*). Therefore, we next determined the effects of TAZ on the regulation of gluconeogenic and inflammatory genes by Dex in mouse liver. TAZ overexpression inhibited the Dex-induced increases in blood glucose and gluconeogenic gene expression, but the effects of TAZ on the inhibition of inflammatory gene expression by Dex, including the expression of the tumor necrosis factor alpha (*Tnfα*) and interleukin 1 (*Il1*) genes, were inconsistent (*Figure 6—figure supplement 2A and B*). This inconsistency may be the result of the ability of TAZ to directly induce hepatic inflammation (*Wang et al., 2016*).

To determine whether the TAZ WW domain is required for GR regulation in vivo, we expressed TAZΔWW, TAZS89A, or GFP in mouse liver. AdTAZΔWW administration had no effects on mouse body and liver mass (*Figure 7—figure supplement 1A and B*). Compared with control AdGFP, TAZS89A reduced the blood glucose of ad libitum-fed, fasted, or pyruvate-challenged mice (*Figure 7A and B*). By contrast, TAZΔWW failed to reduce blood glucose and glucose production during PTT (*Figure 7A*

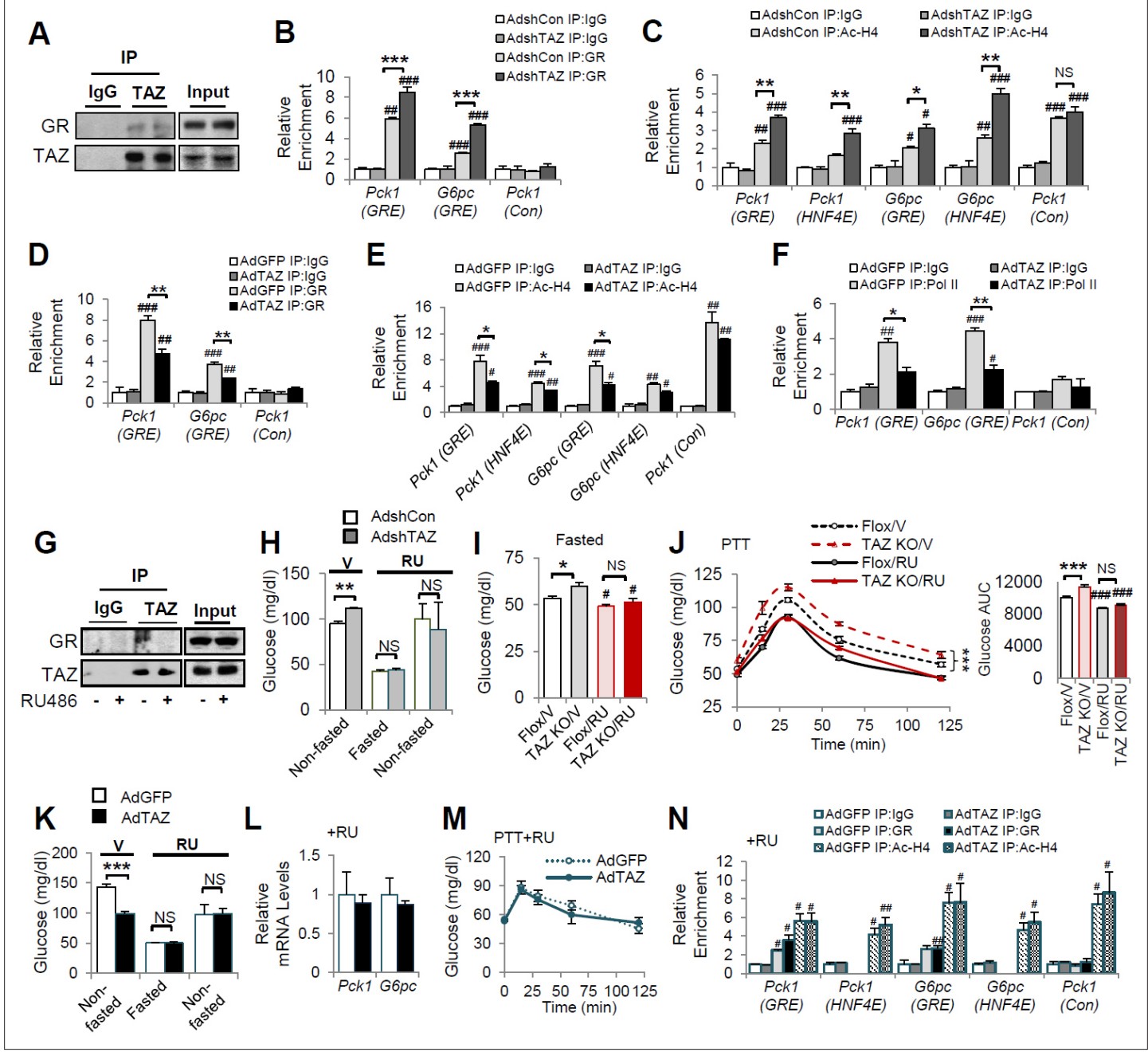

**Figure 6.** TAZ represses glucocorticoid receptor (GR) transactivation of gluconeogenic genes in mouse liver. (**A, G**) Endogenous TAZ was immunoprecipitated from liver nuclear extracts prepared from ad libitum-fed C57BL/6J mice (**A**) or mice treated with RU486 or vehicle (**G**), and the amounts of GR in the immunoprecipitates were measured by immunoblotting. C57BL/6J mice were administered AdshTAZ or AdshCon, as in *Figure 2* (**B, C, H**) or AdTAZ or AdGFP, as in *Figure 3* (**D–F** and **K–N**). (**I, J**) L-TAZ KO and control mice. (**H–N**) Mice were treated with RU486 (RU) or vehicle (V), as indicated. The relative enrichment of GR (**B, D, N**), acetylated-histone 4 (Ac-H4) (**C, E, N**), and Pol II (**F**) in the liver extracts was assessed using chromatin immunoprecipitation (ChIP) assays. (**H, I, K**) Blood glucose concentration. (**L**) Hepatic gene expression. (**J, M**) Pyruvate tolerance testing (PTT). Data are means and SEMs; n = 5–8; for ChIP, the data are the results of triplicate or quadruplicate immunoprecipitations. Data were analyzed by two-way ANOVA (**B–F, I, J, M, N**) and unpaired Student's *t*-test (**H, K, L**). *p<0.05, **p<0.01, ***p<0.001, #p<0.05, ##p<0.01, ###p<0.001; in (**B–F, N**), # denotes comparisons with IgG; in (**I, J**), # denotes comparisons with vehicle-treated controls of the same genotype; NS, not significant.

The online version of this article includes the following figure supplement(s) for figure 6:

**Figure supplement 1.** TAZ does not bind to gluconeogenic gene promoters.

**Figure supplement 2.** Blood glucose concentrations and hepatic gene expression of mice administered AdTAZ and dexamethasone (Dex).

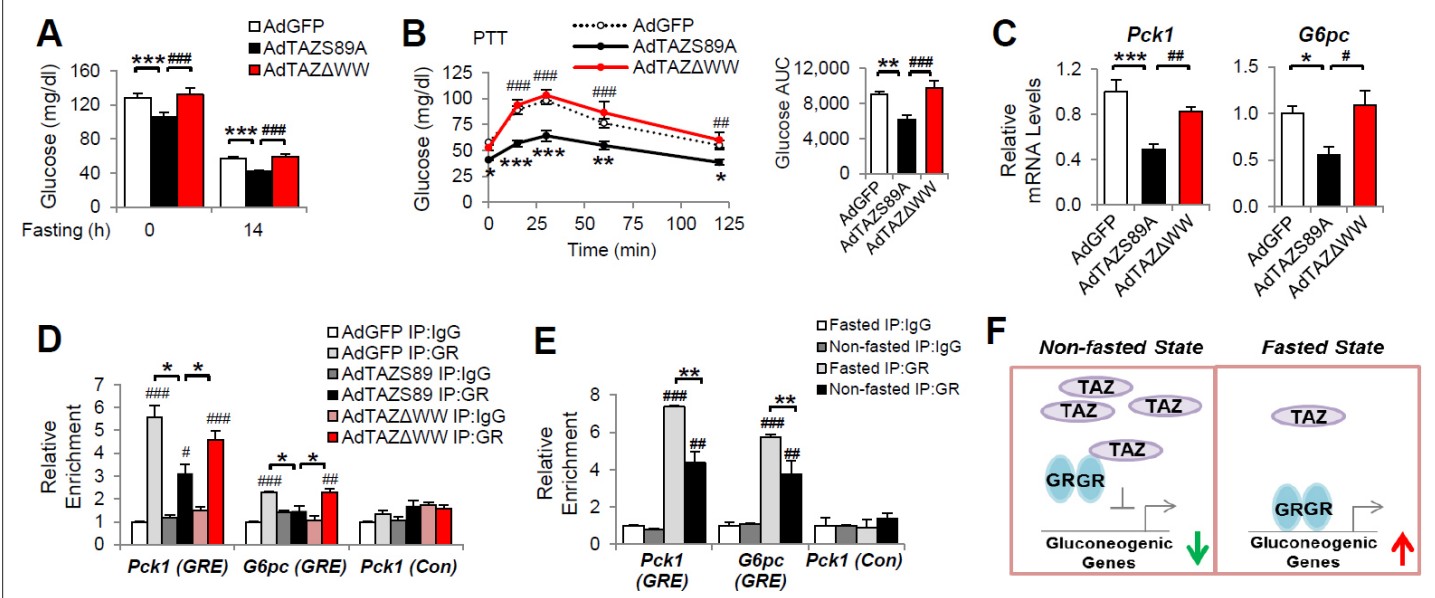

**Figure 7.** The inhibition of gluconeogenic gene expression by TAZ requires its WW domain. C57BL/6J mice (A–D) or (E) C57BL/6J mice were ad libitum-fed (non-fasted) or fasted for 24 hr. (A) Blood glucose. (B) Pyruvate tolerance testing (PTT). (C) Hepatic gene expression. (D, E) Chromatin immunoprecipitation (ChIP) assays were conducted using the indicated antibodies. (F) Regulation of gluconeogenic gene expression by TAZ in the fed and fasting states. Data are means and SEMs; n = 5–8; except for (D) and (E), in which the data are the results of triplicate or quadruplicate immunoprecipitations. Data were analyzed by one-way ANOVA (A, B [right], C), two-way ANOVA (B [left], D, E). *p<0.05, **p<0.01, ***p<0.001, #p<0.05, ##p<0.01, ###p<0.001. In (B), * denotes comparisons with AdGFP and # denotes comparisons between AdS89A and AdTAZΔWW; in (D, E), # denotes comparisons with IgG.

The online version of this article includes the following figure supplement(s) for figure 7:

**Figure supplement 1.** Body and liver mass and hepatic TAZ expression in mice administered AdTAZ mutants or AdGFP.

---

and B), although the TAZΔWW mutant appeared to be more stable in mouse liver (*Figure 7—figure supplement 1C*). In addition, the TAZΔWW mutant failed to inhibit *Pck1* and *G6pc* expression and the binding of GR to GREs in the *Pck1* and *G6pc* promoters in mouse liver (*Figure 7C and D*). Taken together, these data suggest that the WW domain mediates the inhibition of gluconeogenic gene expression by TAZ in vivo.

Fasting increases GR binding to gluconeogenic gene promoters in mouse liver and feeding reduces this binding (*Kalvisa et al., 2018*). Consistent with this, we found that the binding of GR to GREs in the *G6pc* and *Pck1* promoters was increased by >50% in the livers of mice that were fasted versus those that were fed (*Figure 7E*). Collectively, our data suggest a role for hepatic TAZ in glucose homeostasis in normal mouse liver. In the fed state, high hepatic TAZ expression inhibits GR transactivation of gluconeogenic genes by interacting with GR and reducing the binding of GR to the promoters of these genes; whereas in the fasted state, lower TAZ expression enables GR to bind to gluconeogenic genes and activate their transcription, which increases glucose production (*Figure 7F*).

To understand the role of hepatic TAZ in pathophysiology, we used liver-specific IRS 1 and 2 double knockout (L-DKO) mice, a model of diabetes and hepatic insulin resistance. TAZ has previously been shown to regulate IRS2 expression in liver cancer (*Jeong et al., 2018*) and to alter insulin sensitivity (*El Ouarrat et al., 2020*), but we did not observe effects of TAZ on IRS1/2 expression or insulin sensitivity in mice (*Figure 2—figure supplement 5*). TAZ protein expression was reduced in L-DKO mouse liver (*Figure 8—figure supplement 1A*). AdTAZS89A administration affected mouse body and liver mass to a similar extent in both L-DKO and floxed controls (*Figure 8—figure supplement 1B and C*), and importantly, AdTAZS89A reduced blood glucose concentration, glucose production, and gluconeogenic gene expression in both L-DKO and floxed control mice (*Figure 8A–C*), suggesting that TAZ overexpression is sufficient to reduce glucose production in these insulin-resistant mice. In addition, we also measured hepatic TAZ protein expression and investigated the role of TAZ in glucose

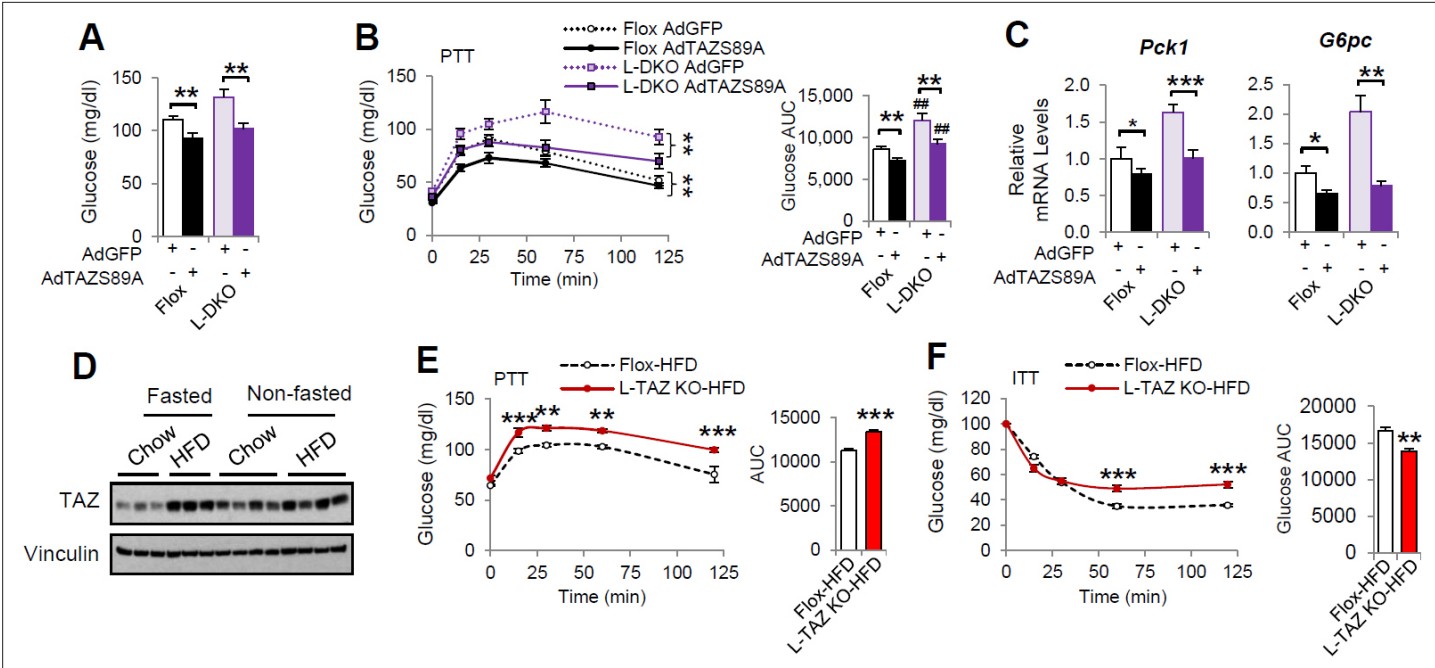

**Figure 8.** TAZ regulates glucose production in insulin-resistant states. (**A–C**) 8- to 10-week-old L-DKO or flox controls were administered the indicated adenoviruses. (**A**) Blood glucose. (**B**) Pyruvate tolerance testing (PTT). (**C**) Hepatic gene expression. (**D**) 4- to 6-week-old C57BL/6J mice were fed chow or a high-fat diet (HFD) for 2 months. Mice that were ad libitum-fed or fasted for 24 hr were compared, and hepatic TAZ expression was measured by immunoblotting whole-cell lysates. (**E, F**) 4- to 6-week-old L-TAZ KO and floxed control littermates were fed an HFD for 2 months. Mice underwent PTT (**E**, left) or insulin tolerance test (ITT) (**F**, left) and the areas under the curves were calculated (right). Data are means and SEMs; n = 5–8. Data were analyzed by unpaired Student's t-test (**A, C, E** [right], **F** [right]) or two-way ANOVA (**B**, **E** [left], **F** [left]); *p<0.05, **p<0.01, ***p<0.001; in (**B**), # denotes comparisons with flox mice administered the same virus.

The online version of this article includes the following figure supplement(s) for figure 8:

**Figure supplement 1.** L-DKO and flox mice administered AdTAZS89A.

production in high-fat diet (HFD)-fed obese mice. HFD increased hepatic TAZ protein level in fasted mice, and to a lesser extent in fed mice, and that L-TAZ KO in mice significantly increased glucose production after pyruvate challenge (*Figure 8D and E*). Consistent with this, L-TAZ KO worsened insulin intolerance in these HFD-fed mice (*Figure 8F*). Taken together, these data suggest that hepatic TAZ regulates glucose homeostasis in insulin resistance.

## Discussion

Distinct cellular functions of YAP and TAZ have been suggested by several studies (*Piccolo et al., 2014*; *Plouffe et al., 2018*; *Morin-Kensicki et al., 2006*; *Makita et al., 2008*). In the liver, YAP is expressed at low levels in normal hepatocytes, which is crucial to the maintenance of their differentiated state (*Yimlamai et al., 2014*). Hepatic YAP expression negatively correlates with the expression of gluconeogenic genes in human hepatocellular carcinoma, which suggests a role for YAP in the integration of glucose metabolic regulation and cell growth (*Hu et al., 2017*). By contrast, TAZ is expressed in normal pericentral hepatocytes and regulates normal glucose homeostasis via GR. The WW domain of TAZ is required for its interaction with GR because a TAZ mutant lacking this domain cannot interact with GR and does not inhibit the transactivation of gluconeogenic genes by GR. Interestingly, although YAP also contains WW domains, it does not interact with GR or a GR co-activator, PGC1α (*Hu et al., 2017*). Therefore, these results demonstrate the distinct roles of TAZ and YAP in the modulation of GR activity and hepatic physiology.

The essential roles of GR and GCs in the promotion of hepatic gluconeogenic gene transcription are well established. Hepatocyte-specific GR ablation leads to the death of half of newborn mice and severe hypoglycemia in adult mice because of a defect in gluconeogenesis (*Opherk et al., 2004*). In addition, GR is required for the full induction of glucose production because fasting, glucagon, cyclic

AMP (cAMP), and epinephrine-activated glucose production are substantially blunted in adrenalectomized rodents (*Exton et al., 1972*; *Winternitz et al., 1957*). The binding of GR to the promoters of gluconeogenic genes is higher in the fasting state than in the fed state (*Kalvisa et al., 2018*), and GCs are believed to play a role in this dynamic regulation. However, little is known about the GC-independent factor(s) or mechanism(s) mediating this regulation. Our data reveal hepatic TAZ to be a novel regulator of GR that modulates the binding of GR to gluconeogenic gene promoters. First, TAZ inhibits GR transactivation. The overexpression of TAZ inhibits, while the knockdown of TAZ increases, the promoter activity of *GRE*-Luc, *G6PC*-Luc, and *PCK1*-Luc reporter genes. In addition, TAZ inhibits the induction of gluconeogenic genes by Dex, a GR agonist. Second, TAZ acts as a repressor of GR. However, unlike a classical co-repressor of GR, such as nuclear receptor compressor 1 (NCoR1), which does not alter DNA-binding by GR (*Patel et al., 2014*), the binding of TAZ to GR results in the dissociation of GR itself from the promoters of GR target genes. Consistent with this, the overexpression of TAZ reduces GR binding to GREs in the promoters of gluconeogenic genes, whereas the knockdown of TAZ increases GR binding to these GREs. Third, the effects of TAZ overexpression or knockdown on gluconeogenic gene expression and the binding of GR to the *Pck1* and *G6pc* promoters are completely abolished in hepatocytes in the presence of RU486, which prevents the interaction of GR with TAZ.

RU486, a GR antagonist, binds to GR but elicits conformational changes that do not favor the recruitment of histone acetyltransferase without inhibition of GR nuclear localization and DNA-binding (*Schulz et al., 2002*). Thus, one explanation for the reduction in the interaction between GR and TAZ following RU486 treatment is that RU486-bound GR possesses a conformation that does not permit GR to interact with TAZ. The ligand-binding domain of GR is crucial for the interaction of GR with co-regulators and ligand binding, but also for the formation of homodimers, which permit GR to bind to GREs in gluconeogenic gene promoters. Our results suggest that the binding of TAZ to the ligand-binding domain of GR does not lead to a reduction in the nuclear localization of GR or in its ability to respond to an agonist; however, exactly how the binding of TAZ to GR causes GR to dissociate from GRE is unknown. GRβ, which is produced by alternative splicing, differs only at its C-terminus from GRα, where the I/LPxY motif resides. Interestingly, unlike GRα, GRβ is not able to bind to conventional GREs in gluconeogenic gene promoters and activate gluconeogenic gene transcription (*Oakley et al., 1996*), which implies that these C-terminal amino acids are indispensable for the binding of GR to GREs.

GRα transactivation of gluconeogenic genes requires the binding of dimeric GR to GREs in the promoters of these genes; however, the transrepression of target genes with anti-inflammatory effects is mediated by monomeric GRα tethered to DNA-bound proinflammatory transcription factors (*Reichardt et al., 2001*). We found that the overexpression of TAZ was sufficient to inhibit the Dex-induced increase in blood glucose and expression of gluconeogenic genes in mouse liver; however, the effects of TAZ on the GR-induced suppression of anti-inflammatory genes were inconsistent. This inconsistency may be the result of the ability of TAZ to directly induce hepatic inflammation (*Wang et al., 2016*). In addition, GR activation induces the transcription of genes involved in glycogen metabolism in a similar manner as gluconeogenic genes; thus, we would expect to find the inhibition of these glycogen metabolic genes by hepatic TAZ. Our data also suggest that the inhibitory effects of TAZ on gluconeogenic gene expression can be dissociated from its effects to promote cell proliferation since TAZS51A and TAZWW mutants cannot activate TEAD-induced proliferative genes (*Zhang et al., 2009*; *Ota and Sasaki, 2008*) but are able to inhibit GR transactivation.

The inhibition of gluconeogenic gene expression by TAZ is hepatocyte-autonomous and is indepedent of glucagon or insulin signaling. TAZ does not affect the phosphorylation of CREB in hepatocytes, nor are plasma glucagon concentrations altered when TAZ is overexpressed or knocked down. Previous studies show that the overexpression of TAZ increases insulin signaling in liver cancer and that TAZ deletion in white adipose and muscle increases insulin sensitivity (*Lei et al., 2008*; *Jeong et al., 2018*). However, we found little effect of either TAZ overexpression or knockdown on the phosphorylation of AKT or FoxO1 (Ser256), or on the plasma insulin concentration, and we found little effects of hepatic TAZ knockout on whole-body insulin sensitivity. In addition, the overexpression of hepatic TAZ is sufficient to reduce blood glucose, pyruvate tolerance, and the hepatic expression of *Pck1* and *G6pc* in liver-specific IRS 1 and 2 double-knockout mice (*Dong et al., 2006*), which strongly

suggests that the inhibition of the GR transactivation of gluconeogenic genes by TAZ is independent of hepatic insulin signaling.

TAZ protein, but not mRNA expression, was altered by fasting and feeding, which suggests that TAZ is subject to post-transcriptional regulation. It is also noteworthy that TAZ is zonally distributed, with the highest expression being near the central veins and the lowest near the hepatic veins. This pattern of distribution is negatively related to the expression and activity of gluconeogenic genes (*Jungermann and Thurman, 1992*), which is consistent with an inhibitory effect of TAZ on hepatic gluconeogenesis under normal physiological conditions. However, the molecular mechanism(s) that mediate the post-transcriptional effects and zonal distribution of TAZ are unclear.

Gluconeogenesis is substantially upregulated in diabetic patients and contributes significantly to the lack of control of blood glucose concentration in these patients (*Magnusson et al., 1992*; *Gastaldelli et al., 2000*). Long-term increases in the concentration and/or actions of endogenous GCs in both humans and rodents manifest as metabolic syndrome, which is characterized by higher glucose production and insulin resistance, and resembles Cushing's syndrome (*Vegiopoulos and Herzig, 2007*). GCs are among the most widely prescribed anti-inflammatory and immunosuppressive drugs (*Overman et al., 2013*). However, GC therapy is associated with substantial increases in glucose production and insulin resistance; thus, therapies that would have the anti-inflammatory effects of GCs, but not affect gluconeogenesis, would be of great interest (*Rosen and Miner, 2005*). Similarly, strategies aimed at specifically inhibiting the ability of GR to induce gluconeogenesis have been explored for the treatment of diabetes (*Jacobson et al., 2005*). However, whether the inhibition of GR by TAZ plays a role in the regulation of hepatic gluconeogenesis in some or all of these abnormal states, and whether a non-tumorigenic TAZ mutant(s), or small peptide(s) or molecule(s), which would mimic or enhance the TAZ-GR interaction could normalize the hyperglycemia associated with obesity, insulin resistance, or the chronic use of GCs, should be determined in future investigations.

In summary, the factors and mechanisms that regulate cell proliferation substantially, such as the mTOR complex (*Saxton and Sabatini, 2017*), overlap with those that control metabolic homeostasis and have emerged as an important area of biology in recent years. We have identified hepatic TAZ, a downstream effector of the Hippo pathway, as a novel regulator of glucose metabolism in the normal liver. TAZ acts as a GR co-repressor and inhibits the binding of GR to GREs in the promoters of gluconeogenic genes, whereby it coordinates GR transactivation and hepatic glucose production in response to fasting and feeding, to maintain energy balance (*Figure 7F*).

# Materials and methods
## Animals and treatments
All mice were fed a standard chow diet and maintained on a 12 hr light/dark cycle, with free access to food and water, unless otherwise indicated, and were sacrificed in the ad libitum-fed, fasted, or re-fed states, as indicated in the main text. At that time, the livers were removed and then fixed in 4% formaldehyde or frozen in liquid nitrogen and stored at −80°C , until RNA, protein, and immunoprecipitation analyses were performed. All animal experiments were performed with the approval of the Institutional Animal Care and Research Advisory Committee at Boston Children's Hospital. Both male and female cohorts of the same age were studied, and we found consistent results in each.

To knock down hepatic TAZ, 8- to 12-week-old C57BL/6J  mice were administered adenoviruses expressing shTAZ (AdshTAZ) or control shRNA (targeting human lamin) by retro-orbital injection at a dose of $1–2 \times 10^9$ pfu/mouse. Six days after virus injection, mice were fasted for 14 hr and subjected to a PTT, as previously described (*Miao et al., 2015*). Briefly, mice were injected intraperitoneally with 2 mg/kg pyruvate at time zero, and subsequently, blood glucose concentrations were measured using a glucometer (Contour). Alternatively, 8 days after virus infection, the mice were sacrificed in the ad libitum state at zeitgeber time (ZT) 13 for gene expression and ChIP analysis. To overexpress TAZ in the liver, mice were injected with adenovirus expressing human flag-TAZ (AdTAZ) or control GFP (AdGFP) retro-orbitally at a dose of $0.5–1 \times 10^9$ pfu/mouse. Five days later, the mice were subjected to PTT or glucagon challenge after an overnight fast or sacrificed at ZT 13 after a 24 hr fast. For glucagon challenge experiment, 8- to 12-week-old C57BL/6J  mice were injected with glucagon (250 µg/kg) intraperitoneally at time zero, and subsequently blood glucose concentrations were measured using a glucometer. To treat RU486, mice were intraperitoneally injected with RU486 (50 mg/kg), fasted for

14 hr, subjected to a PTT, and then sacrificed 1 day later. To treat Dex, mice were injected intraperitoneally with Dex (10 µg/kg) daily for 3 days.

L-TAZ KO mice were generated by crossing TAZ flox/flox mice with *Albumin*-Cre mice. Age- and sex-matched knockout and littermate control mice were studied. Mice underwent PTT and RU496 treatment as described in C57BL/6J mice. To induce obesity and insulin resistance, C57BL/6J , L-TAZ KO, or flox control mice were fed an HFD diet (Research Diet, D12451) for 2 months. Additional information on mice used is included in Key resources table.

Sample sizes were determined based on our previous studies and preliminary results, rather than power calculations, as the effect sizes were not known a priori. Mice were randomized into control and treatment groups, and sample numbers are indicated in figure legends. Data collection and analyses were not performed in a blinded manner.

## Histology, immunohistochemistry, and plasma insulin and glucagon measurements

Portions of liver were fixed overnight in 4% paraformaldehyde, embedded in paraffin, and sectioned, and the sections generated were hematoxylin and eosin-stained and their histology was assessed. For TAZ immunohistochemistry, liver sections underwent antigen retrieval in boiling 10 mM sodium citrate buffer (pH = 6) for 10 min and then were blocked with 5% goat serum in PBS for 1 hr, treated with 3% hydrogen peroxide for 30 min at room temperature, and incubated with primary antibodies overnight at 4°C , followed by horseradish peroxidase (HRP)-conjugated goat anti-rabbit IgG secondary antibodies for 1 hr at room temperature. Immunoreactivity was visualized by incubating slices with VIP peroxidase substrate, with cell nuclei being stained blue using methyl green. All images were obtained using an EVOS2 microscope (Life Technologies). Plasma insulin and glucagon were measured using commercial chemiluminescent ELISA kits.

## Plasmid and adenoviral vector constructs

Human cDNAs of wild-type TAZ and mutants containing deletions of the WW or CC domains were cloned into a pcDNA3-Flag vector by PCR, restriction enzyme digestion, and ligation. TAZS89A, TAZS51A, and GR4A mutants were constructed using site-directed mutagenesis. The *GRE*-Luc construct was constructed by annealing two complementary DNAs containing three repeats of a canonical GRE (two inverted hexameric half-site motifs separated by a three-base-pair spacer), followed by ligation into a pGL3 basic vector (Agilent). The human *PCK1* and *G6PC* luciferase constructs (*PCK1*-Luc and *G6PC*-Luc) and the plasmids encoding HNF4α and PGC1α were gifts from Dr. Pere Puigserver (*Yoon et al., 2003*). The GFP-GR construct was a gift from Dr. Alice Wong, and the human GR gene was cloned into a pcDNA3 vector. The pcDNA3-flag-human YAP1 vector was a gift from Dr. Yosef Shaul (*Levy et al., 2008*), the *TEADE*-Luc (8x*GTIIC*-Luc) construct containing eight repeats of the TEAD response element was a gift from Dr. Stefano Piccolo (*Dupont et al., 2011*), and the TEAD1 expression vector was a gift from Dr. Kun-Liang Guan (*Zhao et al., 2008*). The sequences of all the plasmids were confirmed by sequencing. Information on plasmid constructs is also included in Key resources table. shRNAs targeting mouse or human TAZ, LacZ, and human lamin were constructed in a U6 Block-it vector. Adenoviruses expressing these shRNAs were established as per the manufacturer's instructions and as previously reported (*Miao et al., 2015*). The sequences of the shRNAs are shown in Key resources table. Adenoviruses overexpressing flag-tagged human TAZ, TAZΔWW, TAZS89A, or control GFP were constructed using the AdTrack system (*Luo et al., 2007*). All viruses were amplified in 293A cells, purified using cesium chloride gradient centrifugation, and titrated using an endpoint dilution method. The 293A cells were maintained in high glucose (4.5 g/L) Dulbecco's modified Eagle's medium (DMEM) supplemented with 100 units/mL penicillin, 100 units/mL streptomycin, and 10% FBS at 37°C and in a 5% $CO_2$-containing atmosphere in a humidified incubator, and they were free of mycoplasma contamination. Cell culture reagents were purchased from Life Technologies unless otherwise indicated. Information on cell lines used in this project is included in Key resources table.

## Primary hepatocyte studies

Primary mouse hepatocytes were isolated from 8- to 12-week-old male C57BL/6J mice, as described previously, using a two-step collagenase perfusion method (*Miao et al., 2015*), and they were

maintained at 37°C and in a 5% $CO_2$-containing atmosphere in a humidified incubator. Hepatocytes were seeded in collagen-coated 6-well dishes at a density of $5 \times 10^5$ cells/well in M199 medium supplemented with 2 mM glutamine, 100 units/mL penicillin, 100 units/mL streptomycin, and 10% FBS. After 4 hr, the cells were washed with PBS and incubated in fresh medium, in which various treatments were administered.

For adenovirus-mediated overexpression, adenoviruses were added to cell culture media at multiplicity of infection (MOIs) of 5–15 for 24 hr prior to harvest. Alternatively, for adenovirus-mediated knockdown, hepatocytes were incubated with adenoviruses at MOIs of 15–50 for 36–48 hr prior to harvest. For gene expression studies, on the day of harvest the medium was replaced with fresh M199 medium supplemented with 2 mM glutamine, 100 units/mL penicillin, and 100 units/mL streptomycin. Cells were treated with 20 nM glucagon for 3 hr, 100 nM Dex for 6 hr, or vehicle. Alternatively, cells were pretreated with RU486 (10 µM) for 30 min prior to glucagon treatment. For immunoblotting studies, cells were incubated overnight in M199 without FBS and then stimulated with insulin (20 nM) for 10 min or glucagon (20 nM) for 30 min. All cells were harvested at the same time at the conclusion of each experiment.

To measure glucose production, hepatocytes were fasted overnight in low glucose (1 g/L) DMEM media and then incubated in glucose production medium (2 mM L-carnitine and 2 mM pyruvate, without glucose and phenol red) in the presence of glucagon (20 nM). Small aliquots of medium were removed at various time points, and the concentrations of glucose were measured using an Amplex glucose assay kit, as per the manufacturer's instructions.

Pericentral and periportal hepatocytes were isolated as previously described (*Rajvanshi et al., 1998*). Briefly, primary mouse hepatocytes isolated via the two-step collagenase perfusion method were suspended in 1 mL DMEM media and subjected to Percoll gradient centrifugation at 1000× *g* for 30 min. The gradient consists of 1 mL 70% Percoll followed by 3 mL 52%, 4 mL 42%, and 5 mL 30% Percoll. The periportal and pericentral cell layers were removed and washed with PBS prior to immunoblotting.

## Luciferase reporter assay

HepG2 cells were maintained in DMEM supplemented with 100 units/mL penicillin, 100 units/mL streptomycin, and 10% FBS at 37°C and in a 5% $CO_2$-containing atmosphere, and they were free of mycoplasma contamination. Transient transfection was performed using Lipofectamine 2000 (Life Technologies) as per the manufacturer's instructions. Briefly, cells were seeded in 24-well dishes on day 0. On day 1, cells were transfected with 50–100 ng/well of the indicated firefly luciferase reporter, 50–100 ng/well of *Renilla* plasmid as an internal control for transfection efficiency, and 10–300 ng/well of expression vector. Empty vector was added to ensure that the same amount of total DNA was used in each transfection reaction. On day 2, the cells were either harvested or placed in serum-free media, with or without 100 nM Dex, for an additional 16–20 hr, and were harvested on day 3 by scraping into passive cell lysis buffer. Luciferase activity was then measured using a commercial dual luciferase assay kit, after which firefly luciferase activity was normalized to *Renilla* activity and the values for the control groups were set to 1. Additional information on HepG2 cell line and reagents used is included in Key resources table.

## RNA isolation and analysis

RNA was isolated using Trizol reagent, according to the manufacturer's instructions. For quantitative real-time RT-PCR, 1 µg RNA was reverse transcribed using a High-Capacity cDNA Reverse Transcription Kit. The resulting cDNA was amplified and quantified using SYBR Green PCR Master Mix in QuantStudio 6 Flex. Real-time qRT-PCR was performed in duplicate or triplicate, and the values obtained for each sample were normalized to the expression of the reference genes TATA box binding protein (*Tbp*) for in vivo experiments and ribosomal protein large p0 (*Rplp0*) for in vitro experiments. After normalization, the expression in untreated controls was set to 1. The sequences of the primers used in real-time qRT-PCR are listed in Key resources table.

## Immunoblotting

Whole-cell lysates were prepared by collecting cells in lysis buffer 50 mM Tris, pH 7.5, 150 mM NaCl, 1 mM EDTA, 1% NP-40, 0.5% sodium deoxycholate, 1.0% SDS, 2 mM NaF, and 2 mM $Na_3VO_4$,

supplemented with protease and phosphatase inhibitors, sonicating, and then centrifuging at 13,000× *g* for 10 min (*Miao et al., 2015*). Some cells were treated with 1 μM DSP, a cross-linker, for 30 min on ice prior to harvesting. Liver homogenates were prepared by homogenizing liver tissue in lysis buffer (*Miao et al., 2015*). Nuclear extracts were prepared using an NE-PER extraction kit, according to the manufacturer's instructions, or as previously described (*Miao et al., 2006*). Briefly, livers were homogenized in a Dounce homogenizer in hypotonic buffer (15 mM HEPES, pH 7.9, 1.5 mM MgCl$_2$, 10 mM KCl, 0.2% NP-40, 1 mM EDTA, 5% sucrose, and 1 mM dithiothreitol, supplemented with protease and phosphatase inhibitors). The homogenate was layered onto a sucrose cushion buffer (300 mM sucrose, 60 mM KCl, 10 mM Tris-HCl, pH 7.5, 1 mM EDTA, 0.15 mM spermine, 0.5 mM spermidine, and 1 mM dithiothreitol) and centrifuged at 2000× *g* for 2 min.

Lysates were subjected to SDS-PAGE and transferred onto a PVDF membrane. After blocking, blots were incubated overnight with primary antibody, and then secondary antibody conjugated to horseradish peroxidase and chemiluminescent ECL reagents were used to identify specific bands. Band intensities were determined using ImageJ and normalized to the intensity of loading control bands, and the values of controls were set to 1. The antibodies used in this study were obtained from commercial sources and are listed in Key resources table.

The anti-TAZ antibody has been validated by the vendor and in our laboratory (using overexpressed protein and KO studies) and in published articles. All antibodies for gluconeogenic proteins have been validated by the vendors, published articles, and ourselves using fasting and feeding mouse liver samples.

## Co-immunoprecipitation

Because of poor transfection efficiency in HepG2 and primary mouse hepatocytes, co-IP assays were performed in 293A cells after the co-transfection of expression vectors, as indicated in the figures, for 36–48 hr. Cells were lysed in RIPA buffer (50 mM Tris, pH 7.5, 150 mM NaCl, 1 mM EDTA, 0.5% NP40, and 0.5% sodium deoxycholate) supplemented with protease and phosphatase inhibitors, sonicated, and centrifuged at 13,000× *g* for 10 min. The resulting cell lysates were incubated with anti-flag antibody overnight at 4°C . Immunoprecipitates were obtained by adding Protein A/G agarose, washed three times with RIPA buffer containing 0.025–0.05% SDS, and subjected to immunoblotting. Alternatively, IP was performed using nuclear extracts from mouse livers. The antibodies used for IP are listed in Key resources table.

## Chromatin immunoprecipitation

ChIP assays using primary hepatocytes and liver tissues were performed as previously described, with minor modifications (*Hu et al., 2017*; *Miao et al., 2006*). Cells were cross-linked with 1% formaldehyde at room temperature for 15 min, and then the reaction was stopped by the addition of 125 mM glycine for 5 min at room temperature. Cells were washed with PBS, collected in harvest buffer (100 mM Tris-HCl, pH 9.4, and 10 mM DTT), incubated on ice for 10 min, and then centrifuged at 2000× *g* for 5 min. The cells were then sequentially washed with ice-cold PBS, buffer I (0.25% Triton X-100, 10 mM EDTA, 0.5 mM EGTA, and 10 mM HEPES, pH 6.5), and buffer II (200 mM NaCl, 1 mM EDTA, 0.5 mM EGTA, and 10 mM HEPES, pH 6.5). Livers were prepared in a similar manner: livers from 2 to 4 mice were pooled, minced into small pieces, cross-linked with 1% formaldehyde, and homogenized in hypotonic buffer (15 mM HEPES, pH 7.9, 1.5 mM MgCl$_2$, 10 mM KCl, 0.2% NP-40, 1 mM EDTA, 5% sucrose, and 1 mM dithiothreitol) supplemented with protease and phosphatase inhibitor cocktails. The nuclei were isolated by centrifugation of the resulting homogenates laid over a cushion buffer (300 mM sucrose, 60 mM KCl, 10 mM Tris-HCl, pH 7.5, and 1 mM EDTA). The pellets were resuspended in lysis buffer (1% SDS, 10 mM EDTA, and 50 mM Tris-HCl, pH 8.0) supplemented with protease and phosphatase inhibitor cocktails; sonicated to reduce the DNA length to 0.3–1.5 kb; and then centrifuged at 8000× *g* for 1 min at 4°C. The soluble chromatin was diluted 10-fold in dilution buffer (1% Triton X-100, 2 mM EDTA, 150 mM NaCl, and 20 mM Tris-HCl, pH 8.0) and then pre-cleared with protein A/G agarose beads at 4°C for 1 hr (protein A/G agarose was incubated with sheared salmon sperm DNA and washed three times in dilution buffer prior to use). Equal amounts of pre-cleared chromatin were then added to 2–3 μg of antibody overnight. The next day, 25 μL protein A/G agarose beads were added and the incubation was continued for another 2 hr. The beads were collected and then washed in TSE I (0.1% SDS, 1% Triton X-100, 2 mM EDTA, 20 mM Tris-HCl, pH

8.0, and 150 mM NaCl), TSE II (0.1% SDS, 1% Triton X-100, 2 mM EDTA, 20 mM Tris-HCl, pH 8.1, and 500 mM NaCl), and TE buffer (10 mM Tris, pH 8.0, and 1 mM EDTA), and were finally eluted in 1% SDS/0.1 M NaHCO$_3$. Eluates were heated to 65°C for at least 6 hr to reverse the formaldehyde cross-linking and then treated with proteinase K (Life Technologies). DNA fragments were isolated using a DNA purification kit (Qiagen). The immunoprecipitated DNA and 10% of the pre-cleared chromatin (input DNA) were then subjected to real-time PCR using Power SYBR Green PCR Master Mix, in triplicate. For each immunoprecipitate, we calculated the relative enrichment as $2^{-\Delta Ct}$, where $\Delta Ct$ was calculated as the mean Ct value obtained from the immunoprecipitate DNA minus the mean Ct value obtained from the input DNA. The mean relative enrichment of the replicate immunoprecipitates (3–4 per group) was calculated, and the results are expressed in arbitrary units, with the IgG control for each primer pair set to 1. Representative results from 2 to 4 independent experiments are shown. The antibodies used in the ChIP assays and the real-time PCR primers are listed in Key resources table.

## Statistics

Sample sizes were determined based on our previous studies and preliminary results. Data are presented as mean and SEMs of biological repeats. Representative results of 2–5 independent experiments are shown. Data were analyzed by two-tailed unpaired Student's $t$-test or ANOVA for repeated measurements with multiple comparison post-hoc test using GraphPad Prism software.

## Acknowledgements

This project was supported by NIH awards (DK100539 and DK124328) and a Boston Children's Hospital Career Development Award (to JM).

## Additional information

### Funding

| Funder | Grant reference number | Author |
|---|---|---|
| National Institute of Diabetes and Digestive and Kidney Diseases | DK100539 | Ji Miao |
| National Institute of Diabetes and Digestive and Kidney Diseases | DK124328 | Ji Miao |
| Boston Children's Hospital | Career Development Award | Ji Miao |

The funders had no role in study design, data collection and interpretation, or the decision to submit the work for publication.

### Author contributions

Simiao Xu, Conceptualization, Investigation, Writing – original draft, Writing – review and editing, Formal analysis; Yangyang Liu, Conceptualization, Formal analysis, Investigation, Writing – original draft, Writing – review and editing; Ruixiang Hu, Formal analysis, Investigation, Writing – review and editing, Conceptualization, Writing – original draft; Min Wang, Resources, Writing – review and editing; Oliver Stöhr, Yibo Xiong, Li He, Investigation, Writing – review and editing; Liang Chen, Lingyun Zheng, Songjie Cai, Formal analysis, Investigation, Writing – review and editing; Hong Kang, Cunchuan Wang, Formal analysis, Writing – review and editing; Kyle D Copps, Formal analysis, Writing – original draft, Writing – review and editing; Morris F White, Conceptualization, Formal analysis; Ji Miao, Conceptualization, Formal analysis, Funding acquisition, Investigation, Supervision, Validation, Writing – original draft, Writing – review and editing

### Author ORCIDs

Simiao Xu [ID] http://orcid.org/0000-0001-7834-2404
Ji Miao [ID] http://orcid.org/0000-0003-0869-4492

### Ethics

This study was performed in strict accordance with the recommendations in the Guide for the Care and Use of Laboratory Animals of the National Institutes of Health. All animal experiments were performed with the approval of the Institutional Animal Care and Research Advisory Committee at Boston Children's Hospital (protocols 17-07-3413R and 20-07-4200R).

### Decision letter and Author response

Decision letter https://doi.org/10.7554/eLife.57462.sa1
Author response https://doi.org/10.7554/eLife.57462.sa2

---

## Additional files

### Supplementary files

• Transparent reporting form

### Data availability

There are no sequencing or structural data generated in this manuscript. All data generated and analyzed are included in the manuscript.

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

# Appendix 1

## Appendix 1—key resources table

| Reagent type (species) or resource | Designation | Source or reference | Identifiers | Additional information |
|---|---|---|---|---|
| Genetic reagent (*Mus musculus*) | C57BL/6J | Jackson Laboratory | Stock #: 000664 RRID:IMSR_JAX:000664 | |
| Genetic reagent (*M. musculus*) | $Taz^{flox/flox}$: $Yap^{flox/flox}$ | Jackson Laboratory | Stock #: 030532 RRID:IMSR_JAX:030532 | |
| Genetic reagent (*M. musculus*) | *Albumin*-Cre | Jackson Laboratory | Stock #: 003574 RRID:IMSR_JAX:003574 | |
| Genetic reagent (*M. musculus*) | $Taz^{flox/flox}$: *Alb*-Cre | This paper | | See 'Animals and treatments' in Materials and methods |
| Cell line (*Homo sapiens*) | HepG2 | ATCC | Cat. #: HB-8065 RRID:CVCL 0027 | |
| Cell line (*H. sapiens*) | 293A | Thermo Fisher Scientific | Cat. #: R70507 RRID:CVCL_6910 | |
| Commercial assay or kit | BLOCK-iT U6 RNAi entry vector kit | Thermo Fisher Scientific | Cat. #: K494500 | |
| Commercial assay or kit | BLOCK-iT U6 adenoviral RNAi expression system | Thermo Fisher Scientific | Cat. #: K494100 | |
| Commercial assay or kit | Stellux Chemiluminscence rodent insulin ELISA kit | Alpco | Cat. #: 80-INSMR-CH01 | |
| Commercial assay or kit | Mouse glucagon ELISA kit | Alpco | Cat. #: 48-GLUHU-E01 | |
| Commercial assay or kit | Dual-luciferase reporter assay kit | Promega | Cat. #: E1960 | |
| Commercial assay or kit | VIP substrate Kit, HRP | Vector Laboratories | Cat. #: SK-4600 RRID:AB_2336848 | PMID:28123024 |
| Commercial assay or kit | Mutagenesis kit | Agilent | Cat. #: 210,519 | |
| Commercial assay or kit | cDNA synthesis kit | Thermo Fisher Scientific | Cat. #: 4368813 | |
| Commercial assay or kit | NE-PER Nuclear and Cytoplasmic Extraction kit | Thermo Fisher Scientific | Cat. #: 78835 | |
| Commercial assay or kit | Amplex glucose oxidase assay kit | Thermo Fisher Scientific | Cat. #: A22189 | |
| Recombinant DNA reagent | pCMV-TOPO TAZ (human) | Addgene | Cat. #: 24809 RRID:Addgene_24809 | PMID:18568018 |
| Recombinant DNA reagent | pcDNA3-flag-TAZ (human) | This paper | | See 'Plasmid and adenoviral vector constructs' in Materials and methods |
| Recombinant DNA reagent | pcDNA3-flag-TAZ S51A (human) | This paper | | See 'Plasmid and adenoviral vector constructs' in Materials and methods |
| Recombinant DNA reagent | pcDNA3-flag-TAZ S89A (human) | This paper | | See 'Plasmid and adenoviral vector constructs' in Materials and methods |
| Recombinant DNA reagent | pCMV-TOPO TAZΔWW (human) | Addgene | Cat. #: 24811 RRID:Addgene_24811 | PMID:18568018 |
| Recombinant DNA reagent | pCMV-TOPO TAZΔCC (human) | Addgene | Cat. #: 24816 RRID:Addgene_24816 | PMID:18568018 |
| Recombinant DNA reagent | pcDNA3-flag-TAZΔWW (human) | This paper | | See 'Plasmid and adenoviral vector constructs' in Materials and methods |
| Recombinant DNA reagent | pcDNA3-flag- TAZΔCC (human) | This paper | | See 'Plasmid and adenoviral vector constructs' in Materials and methods |

*Appendix 1 Continued on next page*

*Appendix 1 Continued*

| Reagent type (species) or resource | Designation | Source or reference | Identifiers | Additional information |
|---|---|---|---|---|
| Recombinant DNA reagent | pcDNA3-flag-TAZ WW | This paper | | See 'Plasmid and adenoviral vector constructs' in Materials and methods |
| Recombinant DNA reagent | pGL3-3XGRE-Luc | This paper | | See 'Plasmid and adenoviral vector constructs' in Materials and methods |
| Recombinant DNA reagent | pGL3-G6PC-Luc (human) | Dr. Pere Puigserver | | |
| Recombinant DNA reagent | pGL3-PCK1-Luc (human) | Dr. Pere Puigserver | | |
| Recombinant DNA reagent | pcDNA3-HNF4 α (mouse) | Dr. Pere Puigserver | | |
| Recombinant DNA reagent | pcDNA3-PGC1α (mouse) | Dr. Pere Puigserver | | |
| Recombinant DNA reagent | pEGFP-GR | Addgene | Cat. #: 47504 RRID:Addgene_47504 | |
| Recombinant DNA reagent | pcDNA3-GR (human) | This paper | | See 'Plasmid and adenoviral vector constructs' in Materials and methods |
| Recombinant DNA reagent | pcDNA3-GR4A (human) | This paper | | See 'Plasmid and adenoviral vector constructs' in Materials and methods |
| Recombinant DNA reagent | 8x*GTIIC*-Luc | Addgene | Cat. #: 34615 RRID:Addgene_34615 | PMID:21654799 |
| Recombinant DNA reagent | pcDNA3-Flag-YAP1 (human) | Addgene | Cat. #: 18881 RRID:Addgene_18881 | PMID:18280240 |
| Recombinant DNA reagent | pRK5-TEAD1 (human) | Addgene | Cat. #: 33109 RRID:Addgene_33109 | PMID:18579750 |
| Recombinant DNA reagent | pAd-Track-CMV-GFP | Addgene | Cat. #: 16405 RRID:Addgene_16405 | PMID:9482916 Construct to establish adenovirus |
| Recombinant DNA reagent | pAd-Track-CMV-Flag-TAZ (human) | This paper | | Construct to establish adenovirus expressing TAZ; see 'Plasmid and adenoviral vector constructs' in Materials and methods |
| Recombinant DNA reagent | pAd-Track-CMV-Flag-TAZΔWW (human) | This paper | | Construct to establish adenovirus expressing TAZΔWW; see 'Plasmid and adenoviral vector constructs' in Materials and methods |
| Recombinant DNA reagent | pAd-Track-CMV-Flag-TAZS89A (human) | This paper | | Construct to establish adenovirus expressing TAZS89A; see 'Plasmid and adenoviral vector constructs' in Materials and methods |
| Recombinant DNA reagent | U6-shLamin (human) | This paper | | Control for shTAZ; see 'Plasmid and adenoviral vector constructs' in Materials and methods |
| Recombinant DNA reagent | U6-shTAZ (human) | This paper | | Construct to knockdown TAZ in HepG2 cells; see 'Plasmid and adenoviral vector constructs' in Materials and methods |
| Recombinant DNA reagent | U6-shLacZ | This paper | | Construct to establish adenovirus expressing shControl; control for shTAZ; see 'Plasmid and adenoviral vector constructs' in Materials and methods |
| Recombinant DNA reagent | U6-shTAZ (mouse) | This paper | | Construct to establish adenovirus expressing shTAZ; see 'Plasmid and adenoviral vector constructs' in Materials and methods |

*Appendix 1 Continued on next page*

*Appendix 1 Continued*

| Reagent type (species) or resource | Designation | Source or reference | Identifiers | Additional information |
|---|---|---|---|---|
| Recombinant DNA reagent | Ad-shLacZ | This paper | | Control adenovirus expressing shLacZ; see 'Plasmid and adenoviral vector constructs' in Materials and methods |
| Recombinant DNA reagent | Ad-shTAZ (mouse) | This paper | | Adenovirus expressing shTAZ; see 'Plasmid and adenoviral vector constructs' in Materials and methods |
| Recombinant DNA reagent | Ad-Track-CMV-GFP | This paper | | Control adenovirus expressing GFP, generated from pAd-Track-CMV-GFP vector; see 'Plasmid and adenoviral vector constructs' in Materials and methods |
| Recombinant DNA reagent | Ad-Track-CMV-flag-TAZ | This paper | | Adenovirus generated from pAd-Track-CMV-TAZ; see 'Plasmid and adenoviral vector constructs' in Materials and methods |
| Recombinant DNA reagent | Ad-Track-CMV-flag-TAZΔWW | This paper | | Adenovirus generated from pAd-Track-CMV-TAZΔWW; see 'Plasmid and adenoviral vector constructs' in Materials and methods |
| Recombinant DNA reagent | Ad-Track-CMV-flag-TAZS89A | This paper | | Adenovirus generated from pAd-Track-CMV-TAZS89A; see 'Plasmid and adenoviral vector constructs' in Materials and methods |
| Sequence-based reagent | shLamin (human) | This paper | | CTGGACTTCCAGAAGAACA |
| Sequence-based reagent | shLacZ | This paper | | CTACACAAATCAGCGATTT |
| Sequence-based reagent | sh*TAZ* (human) | This paper | | GCTCAGATCCTTTCCTCAATG |
| Sequence-based reagent | shTAZ (mouse) | This paper | | GCCAGAGATACTTCCTTAATC |
| Sequence-based reagent | Mouse-*Tbp*-F | This paper | qRT-PCR primer | ACCTTCACCAATGACTCCTATG |
| Sequence-based reagent | Mouse-*Tbp*-R | This paper | qRT-PCR primer | TGACTGCAGCAAATCGCTTGG |
| Sequence-based reagent | Mouse-*Cry61*-F | This paper | qRT-PCR primer | CAAGAAATGCAGCAAGACCA |
| Sequence-based reagent | Mouse-*Cry61*-R | This paper | qRT-PCR primer | GGCCGGTATTTCTTGACACT |
| Sequence-based reagent | Mouse-*Ctgf*-F | This paper | qRT-PCR primer | TCCACCCGAGTTACCAATGA |
| Sequence-based reagent | Mouse-*Ctgf*-R | This paper | qRT-PCR primer | CAAACTTGACAGGCTTGGC |
| Sequence-based reagent | Mouse-*G6pc*-F | This paper | qRT-PCR primer | TGGCTTTTTCTTTCCTCGAA |
| Sequence-based reagent | Mouse-*Pck1*-F | This paper | qRT-PCR primer | TCGGAGACTGGTTCAACCTC |
| Sequence-based reagent | Mouse-*Pck1*-R | This paper | qRT-PCR primer | GAGGGACAGCAGCACCAT |
| Sequence-based reagent | Mouse-*Taz*-F | This paper | qRT-PCR primer | ACAGGTGAAAATTCCGGTCA |
| Sequence-based reagent | Mouse-*Taz*-R | This paper | qRT-PCR primer | GAAGGCAGTCCAGGAAATCA |
| Sequence-based reagent | Mouse-*Yap*-F | This paper | qRT-PCR primer | AAGCCATGACTCAGGATGGA |

*Appendix 1 Continued on next page*

*Appendix 1 Continued*

| Reagent type (species) or resource | Designation | Source or reference | Identifiers | Additional information |
|---|---|---|---|---|
| Sequence-based reagent | Mouse-*Yap*-R | This paper | qRT-PCR primer | GTTCATGGCAAAACGAGGGTC |
| Sequence-based reagent | Mouse-*Ctgf*-ChIP-F | This paper | qChIP primer | TTCCTGGCGAGCTAAAGTGT |
| Sequence-based reagent | Mouse-*Ctgf*-ChIP-R | This paper | qChIP primer | CCTTCCTGCCTCATCAACTC |
| Sequence-based reagent | Mouse-*G6pc*-ChIP-*GRE*-F | This paper | qChIP primer | AGCACTGTCAAGCAGTGTGC |
| Sequence-based reagent | Mouse-*G6pc*-ChIP-*GRE*-F | This paper | qChIP primer | GCAAAACAGGCACACAAAAA |
| Sequence-based reagent | Mouse-*G6pc*-ChIP-*HNF4E*-F | This paper | qChIP primer | CCCTGAACATGTTTGCATCA |
| Sequence-based reagent | Mouse-*G6pc*-ChIP-*HNF4E*-R | This paper | qChIP primer | GTAGGTCAATCCAGCCCTGA |
| Sequence-based reagent | Mouse-*Pck1*-ChIP-*Con*-F | This paper | qChIP primer | TGGGAGACACACATCTTATTCCA |
| Sequence-based reagent | Mouse-*Pck1*-ChIP-*Con*-R | This paper | qChIP primer | GTCCCTCTATAGACTTCCAGCACA |
| Sequence-based reagent | Mouse-*Pck1*-ChIP-*GRE*-F | This paper | qChIP primer | TGCAGCCAGCAACATATGAA |
| Sequence-based reagent | Mouse-*Pck1*-ChIP-*GRE*-F | This paper | qChIP primer | TGATGCAAACTGCAGGCTCT |
| Sequence-based reagent | Mouse-*Pck1*-ChIP-*HNF4E*-F | This paper | qChIP primer | TAAGGCAAGAGCCTGCAGTT |
| Sequence-based reagent | Mouse-*Pck1*-ChIP-*HNF4E*-F | This paper | qChIP primer | AGGCCCCTCTATCAGCCATA |
| Antibody | (Rabbit polyclonal) anti-TAZ | Cell Signaling Technology | Cat. #: 4883 RRID:AB_1904158 | PMID:29533785 IB (1:1000) |
| Antibody | (Rabbit monoclonal) anti-p-TAZ (Ser89) | Cell Signaling Technology | Cat. #: 59971 RRID:AB_2799578 | IB (1:1000) |
| Antibody | (Rabbit polyclonal) anti-AKT | Cell Signaling Technology | Cat. #: 9272 RRID:AB_329827 | PMID:23653460 IB (1:1000) |
| Antibody | (Rabbit polyclonal) anti-p-AKT (Thr308) | Cell Signaling Technology | Cat. #: 9275 RRID:AB_329828 | PMID:23715867 IB (1:1000) |
| Antibody | (Rabbit polyclonal) anti-p-AKT (Ser473) | Abclonal | Cat. #: AP0098 RRID:AB_2770899 | IB (1:1000) |
| Antibody | (Mouse monoclonal) anti-beta-actin | Santa Cruz Biotechnology | Cat. #: sc-47778 RRID:AB_2714189 | PMID:28017329 IB (1:3000) |
| Antibody | (Rabbit monoclonal) anti-CREB | Cell Signaling Technology | Cat. #: 9197 RRID:AB_331277 | PMID:24080368 IB (1:1000) |
| Antibody | (Rabbit polyclonal) anti-p-CREB (Ser133) | Abclonal | Cat. #: AP0333 RRID:AB_2771008 | IB (1:1000) |
| Antibody | (Mouse monoclonal) anti-Flag | Abclonal | Cat. #: AE005 RRID:AB_2770401 | IB (1:10000) IP (1 µg/IP) ChIP (1–2 µg/IP) |
| Antibody | (Rabbit polyclonal) anti-Flag | Cell Signaling Technology | Cat. #: 2368 RRID:AB_2217020 | PMID:25514086 IB (1:1000) |
| Antibody | (Rabbit monoclonal) anti-FoxO1 | Cell Signalling Technology | Cat. #: 2880 RRID:AB_2106495 | PMID:24248465 IB (1:1000) |
| Antibody | (Rabbit monoclonal) anti-p-FoxO1 (Ser256) | Cell Signaling Technology | Cat. #: 84192 RRID:AB_2800035 | PMID:31583122 IB (1:1000) |
| Antibody | (Rabbit polyclonal) anti-G6PC | Abcam | Cat. #: ab83690 RRID:AB_1860503 | PMID:25774555 IB (1:1000) |

*Appendix 1 Continued on next page*

*Appendix 1 Continued*

| Reagent type (species) or resource | Designation | Source or reference | Identifiers | Additional information |
|---|---|---|---|---|
| Antibody | (Mouse monoclonal) anti-GAPDH | Santa Cruz Biotechnology | Cat. #: sc-32233 RRID:AB_627679 | PMID:24105481 IB (1:1000) |
| Antibody | (Mouse monoclonal) anti-GLUL | BD Biosciences | Cat. #: 610517 RRID:AB_397879 | PMID:17120293 IB (1:1000) |
| Antibody | (Rabbit polyclonal) anti-GR | Abclonal | Cat. #: A2164 RRID:AB_2764182 | IB (1:3000) |
| Antibody | Goat polyclonal anti-HMGCR | Santa Cruz Biotechnology | Cat. #: sc-27578 RRID:AB_2118199 | PMID:26824363 IB (1:1000) |
| Antibody | (Rabbit polyclonal) anti-HNF4α | Santa Cruz Biotechnology | Cat. #: sc-8987 RRID:AB_2116913 | PMID:29937200 IB (1:000) |
| Antibody | (Rabbit polyclonal) anti-PGC1α | Abclonal | Cat. #: A12348 RRID:AB_2759191 | IB (1:000) |
| Antibody | (Rabbit monoclonal) anti-Tubulin | Cell Signaling Technology | Cat. #: 2125 RRID:AB_2619646 | PMID:28343940 IB (1:5000) |
| Antibody | (Mouse monoclonal) anti-Vinculin | Santa Cruz Biotechnology | Cat. #: sc-73614 RRID:AB_1131294 | PMID:29017056 IB (1:5000) |
| Antibody | (Rabbit polyclonal) anti-YAP | Cell Signaling Technology | Cat. #: 4912 RRID:AB_2218911 | PMID:28323616 IB (1:000) |
| Antibody | (Mouse polyclonal) anti-IRS1 | Provided by Dr. Morris White | | PMID:29867232 IB (1:000) |
| Antibody | (Mouse polyclonal) anti-IRS2 | Provided by Dr. Morris White | | PMID:29867232 IB (1:000) |
| Antibody | (Rabbit polyclonal) anti-Ac-Histone4 | Abclonal | Cat. #: A15233 RRID:AB_2762128 | ChIP (1 µg/IP) |
| Antibody | (Mouse monoclonal) anti-GR | Santa Cruz Biotechnology | Cat. #: Sc-393232 RRID:AB_2687823 | PMID:28467930 ChIP (1–2 µg/IP) |
| Antibody | (Rabbit polyclonal) anti-TAZ | Abclonal | Cat. #: A8202 RRID:AB_2721146 | ChIP (1–2 µg/IP) IHC (1: 200) |
| Antibody | Goat anti-(Rabbit polyclonal) IgG-HPR | Thermo Fisher Scientific | Cat. #: 31460 RRID:AB_228341 | PMID:24932808 IB (1:5000–20,000) |
| Antibody | Goat anti-(Mouse polyclonal) IgG-HRP | Thermo Fisher Scientific | Cat. #: 31430 RRID:AB_228307 | PMID:10359649 IB (1:5000–20,000) |
| Antibody | Rabbit anti-goat (Rabbit polyclonal) IgG-HRP | Santa Cruz Biotechnology | Cat. #: sc-2768 RRID:AB_656964 | PMID:23970784 IB (1:5000–15,000) |
| Chemical compound, drug | Glucagon | Sigma | Cat. #: G2044 | |
| Chemical compound, drug | RU486 | Sigma | Cat. #: M8046 | |
| Chemical compound, drug | Dexamethasone | Sigma | Cat. #: D4902 | For cell culture studies |
| Chemical compound, drug | Dexamethasone | Sigma | Cat. #: 2915 | For in vivo studies |
| Chemical compound, drug | Sodium pyruvate | Sigma | Cat. #: P5280 | |
| Chemical compound, drug | Protease inhibitor cocktail tablet | Sigma | Cat. #: S8820 | |
| Chemical compound, drug | Phosphatase inhibitor cocktail tablet | Sigma | Cat. #: 4906837001 | |
| Chemical compound, drug | Bovine insulin | Sigma | Cat. #: I0516 | For cell culture studies |
| Chemical compound, drug | Human insulin Humulin R U-100 | Eli Lily | Cat. #: HI-210 | For in vivo studies |

*Appendix 1 Continued on next page*

*Appendix 1 Continued*

| Reagent type (species) or resource | Designation | Source or reference | Identifiers | Additional information |
|---|---|---|---|---|
| Chemical compound, drug | Percoll | Cytiva | Cat. #: 17089109 | |
| Chemical compound, drug | Trizol | Thermo Fisher Scientific | Cat. #: 15596018 | |
| Chemical compound, drug | DSP | Thermo Fisher Scientific | Cat. #: PG82081 | |
| Other | SYBR Green PCR master mix | Bioline | Cat. #: BIO-84050 | |
| Other | Collagen Type I Rat Tail | Corning | Cat. #: 354,236 | |
| Other | Collagenase Type I | Worthington Biochemical Corporation | Cat. #: LS004196 | |
| Other | PVDF membrane | Sigma | Cat. #: IPVH00010 | |
| Other | ECL | Thermo Fisher Scientific | Cat. #: A43841 | |
| Other | Agarose A/G beads | Santa Cruz Biotechnology | Cat. #: sc-2003 RRID:AB_10201400 | PMID:28392145 |
| Other | DMEM cell culture media | Thermo Fisher Scientific | Cat. #: 11965118 | |
| Other | M199 cell culture media | Thermo Fisher Scientific | Cat. #:11043023 | |
| Other | DMEM low glucose cell culture media, no phenol red | Thermo Fisher Scientific | Cat. #:11054020 | |
| Other | Lipofectamine 2000 | Thermo Fisher Scientific | Cat. #: 11668019 | Transfection reagent |

