## [Decision Letter]

**Acceptance summary:**

This is an interesting study examining the physiologic role for the hepatic glucocorticoid receptor's transcriptional co-activator TAZ in the regulation of hepatic glucose production. The studies demonstrate an important role for TAZ in the regulation of hepatic glucose metabolism and will be of broad interest to the readership of *eLife*.

**Decision letter after peer review:**

Thank you for submitting your article "TAZ inhibits GR and coordinates hepatic glucose homeostasis in normal physiologic states" for consideration by *eLife*. Your article has been reviewed by 3 peer reviewers, including David E James as the Reviewing Editor and Reviewer #1, and the evaluation has been overseen by Jonathan Cooper as the Senior Editor. The following individuals involved in review of your submission have agreed to reveal their identity: Catherine Postic (Reviewer #2); Rachel Perry (Reviewer #3).

The reviewers have discussed the reviews with one another and the Reviewing Editor has drafted this decision to help you prepare a revised submission.

As the editors have judged that your manuscript is of interest, but as described below that additional experiments are required before it is published, we would like to draw your attention to changes in our revision policy that we have made in response to COVID-19 (https://elifesciences.org/articles/57162). First, because many researchers have temporarily lost access to the labs, we will give authors as much time as they need to submit revised manuscripts. We are also offering, if you choose, to post the manuscript to bioRxiv (if it is not already there) along with this decision letter and a formal designation that the manuscript is "in revision at eLife". Please let us know if you would like to pursue this option. (If your work is more suitable for medRxiv, you will need to post the preprint yourself, as the mechanisms for us to do so are still in development.)

The reviewers were highly impressed by the biochemical analysis in your manuscript showing a novel role for TAZ in the control of hepatic gene expression by glucocorticoids. The thoroughness and multitude of approaches used was laudable. However, the overall tenor was less enthusiastic when it came to the physiological and pathophysiological implications of the data. This sentiment is embodied in the reviewer's comments attached below.

If you feel that you are able to convincingly overturn these concerns then we would encourage you to resubmit your manuscript. However, this will require addressing most of the critical points raised here.

Summary:

In this study the authors have examined the role of TAZ in regulating hepatic gluconeogenesis. The authors show by genetically manipulating the hepatic levels of TAZ that TAZ acts as a repressor of gluconeogenic gene expression and in parallel this regulates hepatic glucose output either in response to a pyruvate tolerance test or to a bolus of glucagon. They go on to show that these effects are mediated via an interaction between the WW domain in TAZ and the glucocorticoid receptor that impairs the ability of the GR to bind to promoter regions. Intriguingly these effects were not observed with YAP another member of the Hippo pathway. These findings extend the expanding role of TAZ in hepatic metabolism. This is an extremely thorough analysis of the role of TAZ in hepatic metabolism involving a series of in vivo and in vitro studies utilizing different approaches to perturb the expression of hepatic TAZ levels. Much of the biochemistry is convincing and the data are well presented. The referees were less enthusiastic about the physiological implications of the data.

Essential revisions:

1. The effects of TAZ overexpression were much more impressive than its under-expression when looking at PTT. In fact if it weren't for the almost non-existent error bars for each BG measurement on the PTT I would almost doubt there is much of a significant effect of TAZ KO. How do the authors explain this? Is this because the mice were so fasted that hepatic TAZ levels are already so low that further reduction in its expression has little effect? This raises the issue of how physiological the level of overexpression of TAZ was. In Figure 3, if I am to interpret this correctly, the level of TAZ in total liver was 3-fold higher in overexpressing mice than controls whereas in the pericentral regions it was expressed at comparable levels to endogenous. Does this mean that there is much TAZ expression in other parts of the liver where TAZ would normally not be found? This needs to be addressed in the manuscript.

2. It would be important to measure TAZ protein concentrations in liver of diabetic mouse models i.e hyperglycemic (genetic or nutritional models of diabetes and/or insulin resistance). Does TAZ affect the binding of GR to gluconeogenic gene promoters under hyperglycemic/diabetic conditions?

3. Would the overexpression of TAZ prevent the hyperglycemia characteristic of db/db mice for example? It would help determine the potential role of TAZ in pathophysiology.

4. Why do the authors not consider – and discuss – the possibility of regulation of glycogenolysis by TAZ-GR? The reduction in liver mass with TAZ knockdown seems more consistent with promoting glycogenolysis (as glycogen will take up more space/account for more liver mass than gluconeogenic precursors) vs gluconeogenesis.

5. The interpretation of the ITT is questionable: the authors state that there was no difference in insulin sensitivity, but if we calculate the plasma glucose concentrations during the ITT based on the time zero plasma glucose concentrations in the PTT, we would expect plasma glucose in the ITT to drop to ~60 mg/dl in the floxed mice and ~70 mg/dl in the L-TAZ KO animals. Given this degree of hypoglycemia, not only insulin sensitivity, but also hypoglycemia counterregulation (which involves glucocorticoids!) would modulate plasma glucose. Please discuss.

---

## [Author Response]

Essential revisions:1. The effects of TAZ overexpression were much more impressive than its under-expression when looking at PTT. In fact if it weren't for the almost non-existent error bars for each BG measurement on the PTT I would almost doubt there is much of a significant effect of TAZ KO. How do the authors explain this? Is this because the mice were so fasted that hepatic TAZ levels are already so low that further reduction in its expression has little effect? This raises the issue of how physiological the level of overexpression of TAZ was.

During a PTT, the intraperitoneally injected pyruvate bolus elicits a glycemic excursion that reflects hepatic gluconeogenesis. Unlike a glucose tolerance test, in which injected glucose induces an increase in the blood glucose concentration up to several fold relative to the basal glucose concentration, we observed a small rise of the blood glucose concentration in the PTT (from ~50–75 mg/dl at the basal time point to ~100 mg/dl at 30 min after pyruvate injection), and thus much smaller variations within each group. The discrepancy of glucose readings at the basal time point among experiments reflects the length of fasting and when the fast started relative to the dark cycle. To demonstrate the small variation, we have plotted our PTT data using SEM (Author response image 1 A, C, and E) and SD (Author response image 1 D, and F).

**Author response image 1. sa2fig1:** PTT data plotted using SEM and SD. Figures 2G (A), 2L (C), and 2M (E), which are plotted using SEM, are compared with Figures R1B, R1D, and R1F, respectively, which are plotted using SD.

We concluded that lowering hepatic TAZ expression via virus-mediated knockdown or genetic KO increases glucose production based on the following evidence. (i) Author response tables 1-3 contain our raw data of the PTT in mice infected with AdshTAZ or AdshCon ( Author response table 1) and L-TAZ KO and control floxed mice (Author response table 1; Author response table 2). The p values at each time point were determined by a two-way ANOVA after Sidak’s multiple comparison correction. (ii) Two-way ANOVA analyses showed significant differences between AdshTAZ versus AdshCon (p < 0.001) and L-TAZ KO versus floxed controls for both female and male mice (p < 0.001). (iii) Unpaired two-tail Student’s t-test analyses showed that the area under the curve significantly differed between AdshTAZ versus AdshCon (p < 0.001, Figure 2G) and L-TAZ KO versus floxed controls for both female and male mice (p < 0.001) (Figures 2L–M). We have included these statistical results in the figure legends to clarify our results and to support our conclusion (Author response table 3) .

**Author response table 1. sa2table1:** Blood glucose concentrations (mg/dl) of AdshTAZ- and AdshCon-infected mice during the PTT .

		Time (min)				
	Treatment	0	15	30	60	120
	AdshCon	76	91	120	105	73
	AdshCon	73	116	139	119	64
	AdshCon	76	109	100	86	71
	AdshCon	67	93	98	103	72
	AdshCon	63	119	119	104	57
	AdshCon	78	104	120	83	68
	AdshCon	66	96	122	106	67
	AdshTAZ	96	129	161	168	91
	AdshTAZ	79	130	149	114	103
	AdshTAZ	80	122	135	116	105
	AdshTAZ	105	195	195	116	108
	AdshTAZ	86	140	162	146	94
	AdshTAZ	85	130	155	122	89
	AdshTAZ	102	129	127	125	94
Ave	AdshCon	71.3	104.0	116.9	100.9	67.4
	AdshTAZ	90.4	139.3	154.9	129.6	97.7
SEM	AdshCon	2.222	4.220	5.302	4.698	2.103
	AdshTAZ	3.981	9.496	8.316	7.615	2.826
SD	AdshCon	5.880	11.165	14.029	12.429	5.563
	AdshTAZ	10.533	25.124	22.003	20.148	7.477
Two-way ANOVA	*p*	0.0942	0.0002	<0.0001	0.0033	0.0017

**Author response table 2. sa2table2:** Blood glucose concentrations (mg/dl) of female L-TAZ KO and floxed control mice during the PTT.

		Time (min)				
	Treatment	0	15	30	60	120
	Flox (F)	56	71	75	63	34
	Flox (F)	44	85	98	70	60
	Flox (F)	52	79	103	86	51
	Flox (F)	54	89	97	63	52
	Flox (F)	51	82	103	85	47
	Flox (F)	52	89	101	76	53
	Flox (F)	53	87	89	83	47
	Flox (F)	54	81	95	66	59
	L-TAZ KO (F)	58	97	120	86	63
	L-TAZ KO (F)	66	102	114	86	73
	L-TAZ KO (F)	58	89	106	91	59
	L-TAZ KO (F)	57	93	111	83	64
	L-TAZ KO (F)	56	93	108	88	62
	L-TAZ KO (F)	58	96	109	87	53
	L-TAZ KO (F)	57	89	105	96	65
	L-TAZ KO (F)	63	94	108	88	59
	L-TAZ KO (F)	62	92	128	106	63
	L-TAZ KO (F)	62	91	113	102	60
Ave	Flox (F)	52.0	82.9	95.1	74.0	50.4
	L-TAZ KO (F)	59.7	93.4	112.2	91.3	62.1
SEM	Flox (F)	1.268	2.142	3.308	3.464	2.890
	L-TAZ KO (F)	1.044	1.343	2.240	2.399	1.629
SD	Flox (F)	3.586	6.058	9.357	9.798	8.176
	L-TAZ KO (F)	3.302	4.248	70.84	7.587	5.152
Two-way ANOVA	*p*	0.0836	0.0067	<0.0001	<0.001	0.002

**Author response table 3. sa2table3:** Blood glucose concentrations (mg/dl) of male L-TAZ KO and floxed control mice during the PTT.

		Time (min)				
	Treatment	0	15	30	60	120
	Flox (M)	46	65	102	69	52
	Flox (M)	44	84	97	73	51
	Flox (M)	51	89	95	79	56
	Flox (M)	54	78	98	69	57
	Flox (M)	53	74	101	77	60
	Flox (M)	46	82	108	77	56
	Flox (M)	54	75	97	68	55
	Flox (M)	51	89	103	79	58
	L-TAZ KO (M)	56	92	129	85	62
	L-TAZ KO (M)	56	94	118	95	59
	L-TAZ KO (M)	59	97	108	88	60
	L-TAZ KO (M)	66	105	110	102	78
	L-TAZ KO (M)	55	106	110	92	66
	L-TAZ KO (M)	61	93	115	92	68
	L-TAZ KO (M)	58	99	105	87	71
Ave	Flox (M)	49.9	79.5	100.1	73.9	55.6
	L-TAZ KO (M)	58.7	98.0	113.6	91.6	66.3
SEM	Flox (M)	1.407	2.897	1.493	1.663	1.051
	L-TAZ KO (M)	1.443	2.138	3.046	2.170	2.552
SD	Flox (M)	3.980	8.194	4.224	4.704	2.973
	L-TAZ KO (M)	3.817	5.657	8.059	5.740	6.751
Two-way ANOVA	*p*	0.0173	<0.0001	<0.0001	<0.0001	<0.0026

In Figure 3, if I am to interpret this correctly, the level of TAZ in total liver was 3-fold higher in overexpressing mice than controls whereas in the pericentral regions it was expressed at comparable levels to endogenous. Does this mean that there is much TAZ expression in other parts of the liver where TAZ would normally not be found? This needs to be addressed in the manuscript.

We thank the reviewer for pointing out the distribution of overexpressed TAZ in the mouse liver. Figure 1E, Figure 1F, and Figure 1—Figure supplement 2 show that endogenous TAZ protein is primarily expressed in pericentral hepatocytes. However, Figure 3—Figure supplement 1 shows that TAZ was overexpressed not only in pericentral hepatocytes, but also in periportal hepatocytes (which did not endogenously express TAZ). This has been clarified in the revised manuscript.

2. It would be important to measure TAZ protein concentrations in liver of diabetic mouse models i.e hyperglycemic (genetic or nutritional models of diabetes and/or insulin resistance). Does TAZ affect the binding of GR to gluconeogenic gene promoters under hyperglycemic/diabetic conditions?3. Would the overexpression of TAZ prevent the hyperglycemia characteristic of db/db mice for example? It would help determine the potential role of TAZ in pathophysiology.

We agree with the reviewer that it is important to understand how TAZ is regulated in the insulin resistant pathologic state and its role in that state. To address this, we measured expression of hepatic TAZ in the following insulin resistant mouse models: HFD-fed mice, leptin-deficient (ob/ob) mice, and liver-specific insulin receptor substrate 1 and 2 double KO mice (L-DKO, a model of hepatic insulin resistance and hyperglycemia). Hepatic TAZ protein expression was reduced in L-DKO mice (Author response image 2 A) but elevated in ob/ob mice (Author response image 2) . In addition, HFD feeding increased the hepatic TAZ protein level (Figure 8D) and L-TAZ KO in HFD-fed mice increased the glucose concentration after mice were challenged with pyruvate (Figure 8E). Moreover, Figures 8A–C show that overexpression of hepatic TAZ reduced blood glucose concentration, glucose production, and mRNA expression of hepatic gluconeogenic genes in hepatic insulin resistant LDKO mice. Taken together these data suggest that hepatic TAZ regulates glucose production in insulin resistant hyperglycemic mice. However, our data also indicate that regulation of hepatic TAZ is complex in HFD-fed insulin resistant states because HFD feeding increased the hepatic TAZ protein level in fasted mice, but to a lesser extent in fed mice (Figure 8D).

**Author response image 2. sa2fig2:** Hepatic TAZ protein expression in hyperglycemic and insulin resistant mouse models. Hepatic TAZ protein expression was measured by immunoblotting liver cytoplasmic (Cyto) and nuclear (Nucl) fractions.

4. Why do the authors not consider – and discuss – the possibility of regulation of glycogenolysis by TAZ-GR? The reduction in liver mass with TAZ knockdown seems more consistent with promoting glycogenolysis (as glycogen will take up more space/account for more liver mass than gluconeogenic precursors) vs gluconeogenesis.

We thank the reviewer for pointing this out. We have included data (Figure 4 – Figure supplement 1) and discussed the regulation of glycogen metabolism by TAZ-GR given that GR is a key regulator of glycogen homeostasis.

5. The interpretation of the ITT is questionable: the authors state that there was no difference in insulin sensitivity, but if we calculate the plasma glucose concentrations during the ITT based on the time zero plasma glucose concentrations in the PTT, we would expect plasma glucose in the ITT to drop to ~60 mg/dl in the floxed mice and ~70 mg/dl in the L-TAZ KO animals. Given this degree of hypoglycemia, not only insulin sensitivity, but also hypoglycemia counterregulation (which involves glucocorticoids!) would modulate plasma glucose. Please discuss.

We thank the reviewer for pointing this out. We have added a new figure (Figure 2—Figure supplement 5C) plotting raw blood glucose concentrations. Original Figure 2—figure supplement 5B plotted glucose concentrations normalized to that at time zero as percentages. New Figure 5C shows that the glucose concentration in L-TAZ KO mice was increased at all time points throughout the ITT; the curve of L-TAZ KO mice paralleled that of control floxed mice. This suggests that the reduced glucose concentrations are primarily due to low basal glucose levels, not alterations in insulin sensitivity. Therefore, we concluded that L-TAZ KO does not significantly alter the insulin sensitivity of mice.

In addition, TAZ KO or overexpression did not significantly affect the plasma insulin or glucagon level (Figure 2—Figure supplement 3A and Figure 3—Figure supplement 2F) or activation of key components/targets of the insulin and glucagon signaling pathways in hepatocytes in vitro and in vivo (Figure 2—Figure supplement 2, Figure 3—Figure supplement 2F, and Figure 4—Figure supplement 3). Therefore, we conclude that regulation of glucose production by TAZ is unlikely due to direct regulation of insulin sensitivity or insulin/glucose signaling pathways. In addition, TAZ overexpression reduced blood glucose concentration, glucose production, and mRNA expression of gluconeogenic genes in L-DKO mice (Figures 8A–C), suggesting that TAZ inhibits gluconeogenesis independent of insulin signaling pathway. However, we agree with the reviewer that although our data show that TAZ inhibits the induction of gluconeogenic gene expression by a GR agonist, dexamethasone, the changes in glucose concentrations induced by hepatic TAZ overexpression or KO may lead to hormonal or non-hormonal compensatory regulation in the liver or extra-hepatic tissues.